# INDEPENDENCE TESTS FOR LANGUAGE MODELS

## ABSTRACT

We consider the following problem of model provenance: can a third party verify whether two language models are trained independently or not given the weights of both models? We propose a family of statistical tests for models of any architecture that yield exact p-values with respect to the null hypothesis that the models are trained with independent randomness (e.g., independent random initialization). These p-values are valid regardless of the composition of either model's training data, and we obtain them by simulating independent copies of each model and comparing various measures of similarity in the weights and activations of the original two models to these independent copies. We evaluate the power of these tests on pairs of 21 open-weight models (210 total pairs) and find they reliably identify all 69 pairs of fine-tuned models. Notably, our tests remain effective even after substantial fine-tuning; we accurately detect dependence between Llama 2 and Llemma, even though the latter was fine-tuned on an 750B additional tokens (37.5% of the original Llama 2 training budget). Finally, we identify transformations of model weights that break the effectiveness of our tests without altering model outputs, and—motivated by the existence of these evasion attacks—we propose a mechanism for matching hidden activations between the MLP layers of two models that is robust to these transformations. Though we no longer obtain exact p-values from this mechanism, empirically we find it reliably distinguishes fine-tuned models and pruned models of different dimension and is even robust to completely retraining the MLP layers from scratch.

## 1 INTRODUCTION

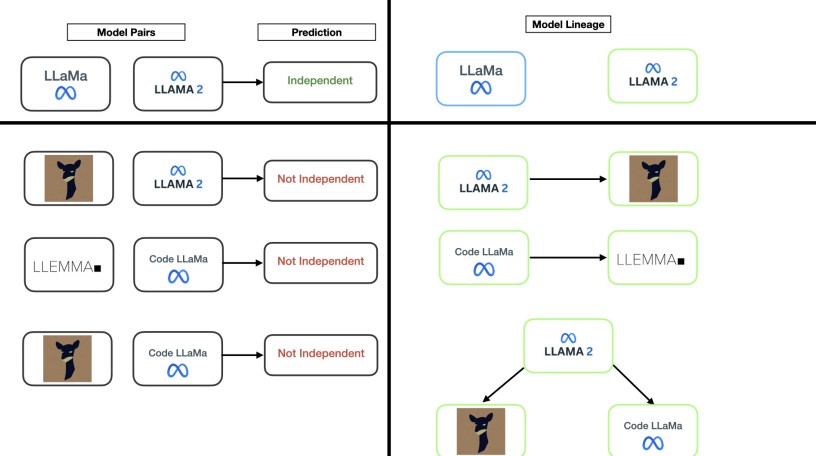

Figure 1: Left: we give examples of model pairs with the Llama architecture and test for independence without knowledge of their origin. Right: we present the ground truth model lineage and highlight distinct cases.

Consider the ways in which two language models may be related (Figure 1): one might be a fine-tune of the other, or they might share a common ancestor, or they may be fully independent. Without knowing the details of either model's training process, what can a third party infer about this relationship from just the weights of both models? Answering this question would better enable independent third parties to track provenance of open-weight models, which is especially important in light of intensifying concerns around intellectual property (IP) protection (Mensch, 2024) and regulatory scrutiny (Anderljung et al., 2023) as both the capabilities and

development costs of these models continue to grow (e.g., (Dubey et al., 2024; Team, 2024; GLM et al., 2024; Yang et al., 2024)).

In this paper, we focus specifically on the question of whether two models are independently trained versus not; we do not distinguish whether one is a fine-tune of the other or if the two models share a common ancestor. Casting the training of a language model as a randomized process (e.g., due to the initial model weights and batch ordering) we formalize this question as a hypothesis testing problem where the null hypothesis is that the weights of the two models are independent random variables. We seek a solution to this problem that admits tight control over false positives while also reliably distinguishing dependent models regardless of 1) design decisions such as the number of fine-tuning tokens or choice of optimizer and 2) the application of various adversarial evasion attacks, including any transformation of model weights that does not affect model output.

In the non-adversarial setting, we propose a family of exact tests for model independence based on identifying similarities between model weights and hidden activations. The main idea is that we can compute p-values by simulating $T$ identically distributed copies of each model and comparing the value of some test statistic (e.g., cosine similarity of model weights) on each of these resampled pairs with the original model pair, where higher values of the test statistic indicate the two models are related. Crucially, we leverage symmetries in training dynamics to simulate these copies without actually having to rerun the full training process, which would be computationally prohibitive. In particular, because the output of a feedforward neural network is invariant to permuting the indices of its hidden units and therefore training dynamics are (typically) permutation equivariant, we can obtain an exact p-value by simply permuting the hidden units of each model and comparing the rank of the test statistic on the original pair to the permuted pairs. Under the null hypothesis that the original two models are independent, the permuted pairs will be exchangeable with the original pair and thus the normalized rank will be uniformly distributed in $\{1/(T+1), ..., 1\}$, yielding a valid p-value.

We evaluate various test statistics—the most effective of which is cosine similarity over weights and activations—on 21 models of the Llama 2 architecture Touvron et al. (2023), including 12 fine-tunes of Llama 2 and nine independently trained models, obtaining negligibly small p-values for all 69 non-independent model pairs. Notably, our tests retain high power over different fine-tuning methods (e.g., different optimizers) and on models extensively fine-tuned for many tokens from the base model such as Llemma Azerbayev et al. (2024), which was fine-tuned on an additional 750B tokens from Llama 2 (i.e., 37.5% of the Llama 2 training budget). These test statistics apply in principle to any model pair for which there exists a pair of layers sharing a common architecture or even a pair of tensors sharing a common shape; thus, we are able to confirm that the leaked Miqu 70B model from Mistral derives from Llama 2 70B and also identify which layers of Llama 3.1 8B are incorporated into Llama 3.2 1B and 3B.

In the adversarial setting, our exact tests are ineffective since an adversary can easily evade them by randomly permuting the hidden units of their model post-training. Moreover, our exact tests apply only to two models of the same architecture; thus, slight changes to the architecture of either model also break these tests. Motivated by these two shortcomings, we design a more robust test statistic that first aligns the hidden units of two models—which may each have different activation types and hidden dimensions—and and then computes some measure of similarity between the aligned models. Due to the alignment step, we can no longer obtain exact p-values with non-trivial power using this test statistic; however, empirically we find we can still reliably distinguish independent and non-independent model pairs. In particular, we find this test statistic empirically behaves like a p-value in the sense that it is close to uniformly distributed in $[0, 1]$ for independent model pairs (and no smaller than 0.024 across all 141 such pairs we test); meanwhile, it is at most 2.2e-308 (the threshold for numerical underflow for a 64-bit float) for all dependent pairs we test (including those for which we simulate an adversary by retraining entire layers from scratch).

## 2 RELATED WORK

The work most closely related to ours is due to Zeng et al. (2024), who consider a similar problem; they develop various tests to determine whether a model is a fine-tune of another by computing the cosine similarity of the products of certain weight matrices in both models. Their focus is on robustness to simple adversaries: these matrix products are invariant to certain kinds of transformations of model weights that preserve model output, and thus their tests are robust to these same transformations. However, we show by construction in D.1 that there exist other such transformations that break all of their tests. Additionally, unlike Zeng et al. (2024), in the non-robust setting we obtain exact p-values from our tests. Jin et al. (2024a) propose crafting specific queries that are likely to produce different responses among independently trained models; their method does not require access to weights but is incapable of producing exact p-values.

A separate line of work on model fingerprinting (Xu et al., 2024; Zhang et al., 2024; Jin et al., 2024b; Yang & Wu, 2024) aims to plant a secret signal in the weights of a model so that anyone who knows the secret key can detect the fingerprint from query access to the model (or derivatives thereof such as fine-tunes). For example, Xu et al. (2024) propose fingerprinting a model by fine-tuning on some secret random string; fingerprint detection then resolves to prompting a putative fingerprinted model with a prefix of the string. Unlike Xu et al. (2024), we do not intervene on the training process of the models we test; however, we do require access to model weights. The fingerprint of Xu et al. (2024) is only detectable with the secret key yet is easily removable by anyone who knows the key; thus, their work does not enable third-party testing of model provenance as do our methods.

Another separate line of work on text watermarking aims to attribute model-generated text by planting a watermark when sampling text from the model (Christ et al., 2023; Kirchenbauer et al., 2024; Kuditipudi et al., 2024; Aaronson & Kirchner, 2023). Because it intervenes on sampling, text watermarking is inapplicable to open-weight models, which are the focus of both model fingerprinting and our setting.

## 3  METHODS

### 3.1  PROBLEM FORMULATION AND TESTING FRAMEWORK

Let $f : \Theta \times \mathcal{X} \to \mathcal{Y}$ denote a *model* mapping parameters $\theta \in \Theta$ and an input $X \in \mathcal{X}$ to an output $f(X; \theta) \in \mathcal{Y}$. We represent a model training or fine-tuning process as a *learning algorithm* $A : \Theta \to \mathcal{P}(\Theta)$, which takes in an initial parameter $\theta^0 \in \Theta$ (e.g., either a random initialization or, in the case of fine-tuning, a base model) and induces a distribution over final parameters. In the context of deep learning, some examples of sources of randomness in a learning algorithm (aside from initialization) include the ordering of minibatches and the use of dropout. Note that $A$ subsumes the choice of training data; our methods make no assumptions on the training data, so we encapsulate it along with other design decisions in the learning algorithm.

Given two models $\theta_1 \sim A_1(\theta_1^0)$ and $\theta_2 \sim A_2(\theta_2^0)$ with initial parameters $\theta_1^0, \theta_2^0 \sim P$ for some joint distribution $P \in \mathcal{P}(\Theta \times \Theta)$, our goal is to test the null hypothesis

$$H_0 : \theta_1 \perp \theta_2, \tag{1}$$

where $\perp$ denotes statistical independence, or the standard definition of independence of two random variables. One example of a case where $\theta_1$ and $\theta_2$ might not be independent is if $\theta_2$ is fine-tuned from $\theta_1$, since in this case we would have $\theta_2^0 = \theta_1$. Indeed, in practice we expect $H_0$ to obtain whenever $\theta_1$ and $\theta_2$ had independent random initializations, i.e., when $\theta_1^0 \perp \theta_2^0$.

---

**Algorithm 1:** Test for computing p-values (`PERMTEST`)

**Input:** Model weights $\theta_1, \theta_2$
**Parameters:** test statistic $\phi$; equivariant transformation class $\Pi$; sample size $T$
**Output:** p-value $\hat{p} \in (0, 1]$

1 **for** $t \in 1, \dots, T$ **do**
2 $\quad \pi_t \sim \mathrm{Unif}(\Pi)$;
3 $\quad \phi_t \leftarrow \phi(\pi_t(\theta_1), \theta_2)$
4 $\hat{p} \leftarrow \frac{1}{T+1}(1 + \sum_{t=1}^{T} \mathbf{1}\{\phi_t \leq \phi(\theta_1, \theta_2)\})$;
5 **return** $1 - \hat{p}$

---

We describe our testing framework for computing p-values against this null hypothesis in Algorithm 1 (`PERMTEST`), where we simulate $T$ independent copies of a model by applying a collection of transformations to the model weights. The validity of these p-values rests on these transformations satisfying certain assumptions with respect to the learning algorithm and random initialization which produced the original model that we capture in the following two definitions.

**Definition 1.** Let $\Pi \subset \Theta \to \Theta$. A distribution $P \in \mathcal{P}(\Theta)$ is $\Pi$-*invariant* if for $\theta \sim P$ and any $\pi \in \Pi$, the parameters $\theta$ and $\pi(\theta)$ are identically distributed.

**Definition 2.** Let $\Pi \subset \Theta \to \Theta$. Consider any $\pi \in \Pi$ and $\theta^0 \in \Theta$, with $\bar{\theta} \sim A(\theta^0), \theta = \pi(\bar{\theta})$ and $\theta' \sim A(\pi(\theta_0))$. A learning algorithm $A$ is $\Pi$-*equivariant* if and only if $\theta$ and $\theta'$ are identically distributed.

In principle, if we know the learning algorithm $A$ and initialization distribution $P$ that produced $\theta_1$, i.e., $\theta_1 \sim A(\theta_1^0)$ for $\theta_1^0 \sim P$, we could obtain an exact p-value with an arbitrary test statistic by repeating the training

process to obtain $T$ independent copies of $\theta_1$ and comparing the test statistic with the original $\theta_1$ to these independent copies; of course, this would be completely impractical in practice due to computational costs. The main idea underlying PERMTEST is that if all we know is that $A$ is $\Pi$-equivariant and $P$ is a $\Pi$-invariant, then we can simulate an identically distributed copy $\theta_1'$ of $\theta_1$ by letting $\theta_1' = \pi(\theta_1)$ for any $\pi \in \Pi$, which allows us to efficiently compute an exact p-value without actually repeating the training process of $\theta_1$. In effect, Definitions 1 and 2 imply that $\pi$ commutes with $A$: sampling $\theta_1' = \pi(\theta_1)$ for $\theta_1 \sim A(\theta_1^0)$ is equivalent to $\theta_1' \sim A(\pi(\theta_1^0))$. We formalize this intuition in the following theorem and subsequently give a concrete toy example; importantly, the result of the theorem holds (under the null hypothesis) without any assumptions on $\theta_2$, meaning that we can have confidence in our test even if we do not trust the provider of $\theta_2$.

**Theorem 1.** *Let $\phi : \Theta \times \Theta \to \mathbb{R}$ be a test statistic and $\Pi \subset \Theta \to \Theta$ be finite. Let $A$ be a $\Pi$-equivariant learning algorithm and let $P$ be a $\Pi$-invariant distribution. Let $\theta_1, \theta_2 \in \Theta$ be independent random variables, with $\theta_1 \sim A(\theta_1^0)$ for $\theta_1^0 \sim P_1$. Then $\widehat{p} = \text{PERMTEST}(\theta_1, \theta_2)$ is uniformly distributed on $\{\frac{i}{T+1}\}_{i=1}^T$.*

*Proof.* From our assumptions on $A$ and $P$ and the fact that $\{\pi_t\}_{t=1}^T$ are independently drawn, it follows that the collection $\{\pi_t(\theta_1)\}_{t=1}^T$ comprises $T$ independent copies of $\theta_1$. The independence of $\theta_1$ and $\theta_2$ thus implies $\{(\pi_t(\theta_1), \theta_2)\}_{t=1}^T$ comprises $T$ independent copies of $(\theta_1, \theta_2)$, and so the claim follows by symmetry. □

Standard initialization schemes for feedforward networks are symmetric over the hidden units of the network, and so one example of a class of transformations with respect to which any such initialization is invariant is the set of permutations over the hidden units of the network. Moreover, the gradient of the model's output with respect to the hidden units is permutation equivariant; thus, any learning algorithm whose update rule is itself a permutation equivariant function of gradients (e.g., SGD, Adam, etc.) satisfies Definition 2 with respect to these transformations. The following toy example makes these claims concrete by applying PERMTEST to a standard two-layer MLP to obtain an exact p-value.

**Example 1:** Let $\theta = (W_1, W_2) \in \Theta$ parameterize a two-layer $m$ hidden unit MLP with $f(x; \theta) = W_2 \sigma(W_1 X)$, for some element-wise activation function $\sigma : \mathbb{R} \to \mathbb{R}$. Let $(x, y) \in \mathcal{X} \times \mathcal{Y}$ for $\mathcal{X} = \mathbb{R}^d$ and $\mathcal{Y} = \mathbb{R}$ be a training example. Let $P \in \mathcal{P}(\theta)$ be the standard isotropic Gaussian distribution over $\Theta$ with variance $\sigma^2$, and let $A$ denote running standard gradient descent on the loss $L(\theta) = \ell(f(X; \theta), y)$ for some arbitrary $\ell : \mathcal{Y} \times \mathcal{Y} \to \mathbb{R}$. Abusing notation, identify $\Pi$ with the set of $m \times m$ permutation matrices such that for $\pi \in \Pi$ we have $\pi(\theta) = (\pi W_1, W_2 \pi^T)$. Observe $P$ is $\Pi$-invariant and also $A$ is $\Pi$-equivariant irrespective of $\ell$: for any $\pi \in \Pi$ we have $\pi(\nabla_\theta f_\theta(X)) = \nabla_{\pi(\theta)} f_{\pi(\theta)}(X)$, which implies $\nabla_\theta L(\theta)$ is $\Pi$-equivariant. Let $\phi : \Theta \times \Theta \to \mathbb{R}$ be the negative $\ell_2$ distance between the weights of the two models, i.e., $\phi(\theta_1, \theta_2) = -\|\theta_1 - \theta_2\|_2$. Then if $\theta_1 \sim A(\theta_1^0)$ for some $\theta_1^0 \sim P$, for any random variable $\theta_2 \in \Theta$ the output $\widehat{p} = \text{PERMTEST}(\theta_1, \theta_2)$ is a valid p-value. In particular, if $\theta_1$ and $\theta_2$ are independent then $-\|\theta_1 - \theta_2\|_2$ will be identically distributed as $-\|\pi(\theta_1) - \theta_2\|_2$ for any $\pi \in \Pi$. If on the other hand $\theta_2$ is a fine-tune of $\theta_1$, we might expect $-\|\pi(\theta_1) - \theta_2\|_2 \ll -\|\theta_1 - \theta_2\|_2$. ◊

### 3.2 TEST STATISTICS

We have shown PERMTEST produces a valid p-value regardless of the test statistic $\phi$ we use. The sole objective then in designing a test statistic is to achieve high statistical power: we would like $\widehat{p} = \text{PERMTEST}(\theta_1, \theta_2)$ to be small when $\theta_1$ and $\theta_2$ are not independent. The test statistics we introduce in this section apply to any model pair sharing the same architecture. In particular, the test statistics all share the following form based on Algorithm 2 (MATCH): for $m, n \in \mathbb{N}$ and $M : \Theta \to \mathbb{R}^{n \times m}$, let

$$\phi_M(\theta_1, \theta_2) := \rho(\text{MATCH}(M(\theta_1), M(\theta_2)), [1, ..., h]). \tag{2}$$

Equation (2) is applicable to any model architecture $\Theta$ for which we can define a suitable matrix valued function of model parameters. Taking various such functions $M$ yields different test statistics. We focus our experiments on Transformer models consisting of a series of $L$ Transformer blocks that each contain an MLP module, and we take $M(\theta)$ to be either the first-layer (i.e., up projection) weights or the hidden-layer activations of one of these MLP modules. In particular, let $U^{(\ell)}(\theta) \in \mathbb{R}^{h \times d}$ denote the first layer up projection weights of the MLP module in the $\ell$-th block, where $h$ is the hidden dimension and $d$ is the input dimension, and let $H^{(\ell)}(\theta) \in \mathbb{R}^{h \times (N \cdot n)}$ denote the hidden activations that obtain from passing $N$ input sequences of length $n$, $X \in \mathbb{R}^{N \times n \times d}$ to the same MLP module (the test is valid for any $X$; we will specify later how we choose $X$ in our experiments). The two main test statistics we will employ in our experiments are $\phi_{U^{(\ell)}}$ and $\phi_{H^{(\ell)}}$. In both these cases, the idea is

to match the hidden units between the $\ell$-th MLP modules of the two models in a way that maximizes the cosine similarity of the corresponding row pairs and then return the Spearman correlation of this matching with the identity permutation. We describe matching in Algorithm 2, wherein `cossim` denotes cosine similarity function and `LAP` denotes the algorithm of Ramshaw & Tarjan (2012) we use to solve the matching problem.

Both $U^{(\ell)}$ and $H^{(\ell)}$ are equivariant with respect to permuting the hidden units of the corresponding MLP module, and so we can use `PERMTEST` to compute p-values from both $\phi_{U^{(\ell)}}$ and $\phi_{H^{(\ell)}}$ by taking $\Pi$ to be the set of permutations over the hidden units of the $\ell$-th MLP, similar to Example 1. Doing so would require recomputing these test statistics $T$ times to obtain a p-value less than $1/T$. Instead, observe that if $\theta_1 \perp \theta_2$ then letting $\pi = \text{LAP}(C)$ in `MATCH` is equivalent in distribution to sampling $\pi \sim \text{Unif}(\Pi)$. Thus, instead of running `PERMTEST` itself, in our experiments we convert the output $\widehat{\rho}$ to a p-value $\hat{p}$ using `scipy.stats` for the Spearman correlation coefficient (where $\hat{p} = 2\mathbb{P}(T > \widehat{\rho})$ for a t-distribution $T$ with $h - 2$ degrees of freedom). Doing so allows us to obtain an estimate of the exact p-value at a finer scale (as the null distribution of the Spearman correlation coefficient is known) without incurring extra computational costs. We will still employ `PERMTEST` in our experiments to compute p-values with other baselines from prior work (e.g., $\ell_2$ distance between weights), since unlike $\phi_{U^{(\ell)}}$ and $\phi_{H^{(\ell)}}$ the null distribution of these statistics will vary depending on the specific model pair we are testing.

---

**Algorithm 2:** Cosine similarity matching (`MATCH`)

**Input:** Matrices $W_1, W_2$ with $h$ rows
**Output:** Correlation coefficient $\widehat{\rho} \in [-1, 1]$
1 **for** $i \in 1, \ldots, h$ **do**
2     **for** $j \in 1, \ldots, h$ **do**
3        $C_{i,j} \leftarrow \text{cossim}((W_1)_i, (W_2)_j)$;
4 $\pi \leftarrow \text{LAP}(C)$;
5 **return** $\pi$

---

Finally, because $\phi_{U^{(\ell)}}$ and $\phi_{H^{(\ell)}}$ are both functions of only the $\ell$-th block of the model and we can independently permute the hidden units of the MLP in the $\ell$-th block without affecting the inputs or outputs of the other blocks, the p-values we obtain from $\phi_{U^{(\ell)}}$ and $\phi_{H^{(\ell)}}$ are independent across blocks and thus we can aggregate them using Fisher's method (Mosteller & Fisher (1948)) to obtain a more powerful test in Algorithm 3 (`FISHER`).

---

**Algorithm 3:** Aggregating p-values (`FISHER`)

**Input:** Model weights $\theta_1, \theta_2$
**Parameters:** test statistics $\{\phi^{(i)}\}_{i=1}^L$; transformation classes $\{\Pi^{(i)}\}_{i=1}^L$; sample size $T$
**Output:** p-value $\hat{p} \in (0, 1]$
1 **for** $i \in 1, \ldots, L$ **do**
2     $\widehat{p}^{(i)} \leftarrow \text{PERMTEST}(\theta_1, \theta_2; \phi^{(i)}, \Pi^{(i)}, T)$
3 $\xi \leftarrow \sum_{i=1}^L \log \widehat{p}^{(i)}$;
4 $\hat{p} \leftarrow 1 - \mathbb{P}(\chi_{2L}^2 < -2\xi)$;
5 **return** $\hat{p}$

---

**Theorem 2.** *Let $i, j \in [L]$ with $i \neq j$. Suppose for $\ell \in \{i, j\}$ that*

    *1. $M^{(\ell)} : \Theta \to \mathbb{R}^{h \times N}$ is equivariant with respect to $\Pi^{(\ell)}$, i.e., for any $\theta \in \Theta$ and $\pi^{(\ell)} \in \Pi^{(\ell)}$ we have*

$$M(\pi^{(\ell)}(\theta)) = \pi^{(\ell)} M(\theta).$$

    *2. $A$ is a $\Pi^{(\ell)}$-equivariant learning algorithm and $P \in \mathcal{P}(\Theta_{\text{LM}})$ is a $\Pi^{(\ell)}$-invariant distribution.*

*Let $\theta_1, \theta_2 \in \Theta$. If $\theta_1 \perp \theta_2$ for $\theta_1 \sim A(\theta_1^0)$ with $\theta_1^0 \sim P$, then*

$$\text{MATCH}(M^{(i)}(\theta_1), M^{(i)}(\theta_2)) \perp \text{MATCH}(M^{(j)}(\theta_1), M^{(j)}(\theta_2)).$$

*Proof.* Let $\theta_1' \sim A(\pi_1^{(i)} \circ \pi_2^{(j)}(\theta_1^0))$ for $\pi_1, \pi_2 \overset{\text{i.i.d.}}{\sim} \text{Unif}(\Pi)$. Then $\theta_1'$ is an independent copy of $\theta_1$ since taking the composition $\pi_1^{(i)} \circ \pi_2^{(j)}(\theta_1)$ yields an independent copy of $\theta_1$ for any $\pi_1, \pi_2 \in \Pi$. From $\theta_1 \perp \theta_2$, it follows

for $\ell \in \{i, j\}$ that $\texttt{MATCH}(M^{(\ell)}(\theta_1'), M^{(\ell)}(\theta_2))$ is identically distributed to $\texttt{MATCH}(M^{(\ell)}(\theta_1), M^{(\ell)}(\theta_2))$. The result then follows from the fact $\texttt{MATCH}$ is equivariant with respect to permuting the rows of its arguments: in particular, for any $\pi \in \Pi$ we have $\texttt{MATCH}(\pi W_1, W_2) = \pi \texttt{MATCH}(W_1, W_2)$. $\square$

Recall $\phi_{U^{(\ell)}}$ and $\phi_{H^{(\ell)}}$ are functions of $\texttt{MATCH}(M^{(\ell)}(\theta_1), M^{(\ell)}(\theta_2))$ respectively for $M^{(\ell)} = U^{(\ell)}$ and $M^{(\ell)} = H^{(\ell)}$, both of which satisfy the assumptions of the theorem. Thus, the result of the theorem applies to both these test statistics, and the independence of the p-values from these test statistics across blocks follows directly from the independence of the statistics themselves.

### 3.3 ROBUSTNESS TO ADVERSARIAL MANIPULATION

It is easy to fool the tests we have proposed thus far by applying simple transformations to model parameters that do not change model output: in particular, an adversary can fool $\phi_{U^{(\ell)}}$ and $\phi_{H^{(\ell)}}$ by randomly permuting the hidden units of the corresponding MLP.[1] Motivated by this shortcoming, we design a robust test that aligns the hidden layer activations of a particular layer between the two models and evaluates the quality of the alignment. Notably, we can compute the alignment between two layers with different numbers of hidden units; thus, unlike the tests in Section 3.2, we can apply our robust test to any model pair regardless of whether the two models share the same architecture. This statistic is also robust to the transformation described in Appendix D.1 that breaks the invariants proposed by Zeng et al. (2024).

We first describe a test specific to architectures with Gated Linear Units (GLUs) (Dauphin et al., 2016)—a category which includes the Llama model architecture among many other language models—and then describe a general extension of this test to any architecture involving hidden activations. We evaluate both the GLU-specific version and the more general extension in our experiments.

#### 3.3.1 TESTING GLU MODELS

We first consider models of the following architecture (Definition 3).

**Definition 3.** (GLU MLP) Let $G, U \in \mathbb{R}^{h \times d}$ and $D \in \mathbb{R}^{d \times h}$. Let $\sigma : \mathbb{R} \to \mathbb{R}$ be an element-wise activation function. For $X \in \mathbb{R}^d =: \mathcal{X}_{\mathrm{mlp}}$ and $\theta = (G, U, D) \in \Theta_{\mathrm{mlp}}^h$, let $f_{\mathrm{mlp}}(X; \theta) := D(\sigma(GX) \odot (UX))$.

Let $\theta_i = (G_i, U_i, D_i) \in \Theta_{\mathrm{mlp}}^h$ for $i \in \{1, 2\}$, and for some $X \in \mathcal{X}_{\mathrm{mlp}}^N$ let $H_{\mathrm{up}}(\theta_i) = U_i X \in \mathbb{R}^{h \times N \times n}$ be the output of the up projection operation and let $H_{\mathrm{gate}}(\theta_i) = G_i X \in \mathbb{R}^{h \times N \times n}$ be the output of the gate projection operation. Then define the test statistic by

$$\phi_{\mathrm{MATCH}}(\theta_1, \theta_2) := \rho(\texttt{MATCH}(H_{\mathrm{up}}(\theta_1), H_{\mathrm{up}}(\theta_2)), \texttt{MATCH}(H_{\mathrm{gate}}(\theta_1), H_{\mathrm{gate}}(\theta_2))).$$

The main idea of $\phi_{\mathrm{MATCH}}$ is that an adversary who permutes the output of the gate projection operation in some block must also permute the output of the up projection operation in the same way to preserve model output (due to the direct product operation); thus, high correlation between the best alignment of the two models' gate projection activations with that of the up projection activations suggests the two models may not be independent.

#### 3.3.2 BEYOND GLU MODELS

So far we have given a test statistic $\phi_{\mathrm{MATCH}}$ that we can compute for two GLU MLPs of the same width (i.e., number of hidden units). The $\texttt{MATCH}$ algorithm works for two matrices with different column sizes, so we can straightforwardly extend $\phi_{\mathrm{MATCH}}$ to apply to pairs $\theta_1 \in \Theta_{\mathrm{mlp}}^{h_1}$, $\theta_2 \in \Theta_{\mathrm{mlp}}^{h_2}$ with $h_1 \neq h_2$. We can also extend $\phi_{\mathrm{MATCH}}$ to apply to models that call a GLU MLP as a sub-module. In particular, for $\theta \in \Theta_{\mathrm{mlp}}^h$ consider a model of the form $f(x; \theta) = g(\{f_{\mathrm{mlp}}(h_i(x); \theta)\}_{i=1}^n)$ for some $h_i : \mathcal{X} \to \mathcal{X}_{\mathrm{mlp}}$ for $i \in [n]$ and $g : \mathbb{R}^{n \times d} \to \mathcal{Y}$.[2] We can apply $\phi_{\mathrm{MATCH}}$ to such a model by obtaining activation matrices $H_{\mathrm{up}}(\theta)$ and $H_{\mathrm{gate}}(\theta)$ from passing multiple inputs $x \in \mathcal{X}$ to the model $f$ and concatenating over $i \in [n]$. In the case that a model has multiple GLU MLPs, we also use $\texttt{FISHER}$ to aggregate the statistics.

---

[1]If an adversary randomly permutes the hidden units of each MLP layer of their model, then when we run $\texttt{PERMTEST}$ with either $\phi_{U^{(\ell)}}$ or $\phi_{H^{(\ell)}}$ each of the resampled statistics $\{\phi_t\}_{t=1}^T$ will be identically distributed to the original statistic, in which case the output will be uniform on $[0, 1]$.

[2]This family of models includes the Llama 2 architecture, which broadcasts $f_{\mathrm{mlp}}$ across the sequence dimension.

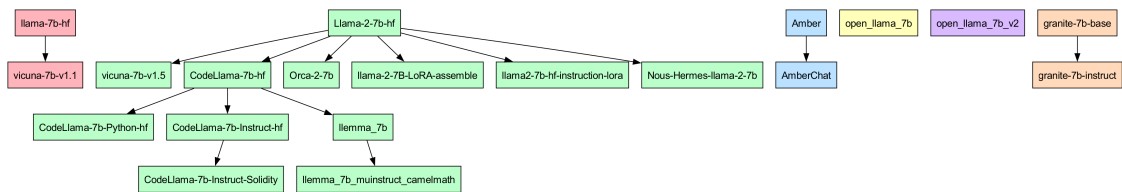

Figure 2: We enumerate the public Llama-7B models and delineate the sets of dependent model pairs by color.

Finally, in principle we can even apply $\phi_{\mathrm{MATCH}}$ to models that do not involve GLU MLPs at all. To this end, consider a model $f(x) = g(\{f(h_i(x))\}_{i=1}^n)$. Though $f$ itself does not involve a GLU MLP, we can first learn parameters $\theta \in \Theta_{\mathrm{mlp}}^h$ to minimize the expected difference between $f(X)$ and $f(X;\theta)$ over some distribution on randomly sampled Gaussian inputs $X \in \mathbb{R}^{N \times n \times d_{\mathrm{embed}}}$ and then apply $\phi_{\mathrm{MATCH}}$ to these learned parameters. Perhaps surprisingly, we show this test is effective in practice at distinguishing independent versus non independent models. See Section 4.2 and Appendix H.3 for details.

## 4 EXPERIMENTAL RESULTS

### 4.1 NON-ADVERSARIAL SETTING

We now validate the effectiveness of our tests against public open-weight language models. We first consider 21 models trained with the Llama-7B architecture with public documentation on ground truth model independence, which we highlight in Figure 2. For these models, with architecture $\Theta_{\mathrm{LM}}$, have a GLU MLP component described in Section 3.3.1 and Appendix A. For $i \in \{1, 2\}$, we define our statistics $\phi_{U^{(\ell)}}$ and $\phi_{H^{(\ell)}}$ as follows. Let $U_i^{(\ell)} \in \mathbb{R}^{h \times d}$ be the up projection matrix in the $\ell$-th block of $\theta_\ell \in \Theta_{\mathrm{LM}}$, and

$$\phi_{U^{(\ell)}}(\theta_1, \theta_2) := \rho(\mathtt{MATCH}(U_1^{(\ell)}, U_2^{(\ell)}), [1, ..., h]),$$

And, let $H^{(\ell)}(\theta_i) \in \mathbb{R}^{h \times N \times n}$ be the input to the down projection operation, such that for $X \in \mathcal{X}^N$ we have $H^{(\ell)}(\theta_i)_{:,j,k} = \sigma(G_i^{(\ell)} X_{\theta_i}^{(\ell-1)}) \odot (U_i^{(\ell)} X_{\theta_i}^{(\ell-1)})$, and let $\phi_{H^{(\ell)}}$ be

$$\phi_{H^{(\ell)}}(\theta_1, \theta_2) := \rho(\mathtt{MATCH}(H_{\theta_1}^{(\ell)}, H_{\theta_2}^{(\ell)}), [1, ..., h]).$$

In addition to $\phi_{U^{(\ell)}}$ or $\phi_{H^{(\ell)}}$, we employ two test statistics from prior work as baselines: Jensen-Shannon divergence between next token output distributions ($\phi_{\mathrm{JSD}}$, Lin (2006)) and $\ell_2$ distance between weights ($\phi_{\ell_2}$, from Xu et al. (2024)). Since the Jensen-Shannon divergence is (by definition) invariant to any transformation of weights that does not affect model output, we cannot compute meaningful p-values using PERMTEST; instead, in our experiments we report the raw value of the test statistic itself. As for $\ell_2$ distance, to be consistent with prior work (Xu et al. (2024)), we define for $\theta_1, \theta_2 \in \Theta_{\mathrm{LM}}$,

$$\phi_{\ell_2}(\theta_1, \theta_2) := -\sum_{i=1}^{L} \ell_2(\theta_1^{(\ell)}, \theta_2^{(\ell)})$$

where $\theta_1^{(\ell)}, \theta_2^{(\ell)}$ are the $\ell$-th layers of $\theta_1, \theta_2$, respectively (we assume their dimensions align). We take the negation due to the design of PERMTEST, in which higher values of the test statistic indicate dependence. We obtain p-values from $\phi_{\ell_2}$ by running PERMTEST with $\Pi$ as the set of permutations over both the hidden units of each MLP and the embedding dimension of the model (i.e., the inputs passed to the both the MLP and self-attention layers in each block); we defer the precise definition of $\Pi$ in this case to Appendix B.

The 21 models we evaluate include 6 base models (trained from scratch), so we have six disjoint sets of the models based on Llama-2-7b-hf stemming from a diverse mix of industry labs and non-profits (Azerbayev et al., 2024; Sudalairaj et al., 2024; Liu et al., 2023; Li et al., 2023). We consider any pair of models in the same tree as dependent and all other pairs as independent. We include examples of further fine-tunes (e.g., llemma_7b) of fine-tunes (e.g., CodeLlama-7b-hf) among the models we test. We will mostly refer to models using by their Huggingface identifiers, without the organization names for clarity.

We evaluate four test statistics: $\phi_{U^{(\ell)}}$ (cosine similarity of weights), $\phi_{H^{(\ell)}}$ (cosine similarity of hidden activations), $\phi_{\ell_2}$ ($\ell_2$ distance), and $\phi_{\mathrm{JSD}}$ (Jensen-Shannon Divergence). As we describe in Section 3.2, for $\phi_{U^{(\ell)}}$ and $\phi_{H^{(\ell)}}$ we report aggregated p-values over all blocks using FISHER. We compute p-values for $\phi_{\ell_2}$ using PERMTEST with $T = 99$, while for $\phi_{\mathrm{JSD}}$ we simply report the raw values of the test statistic itself. We compute $\phi_{\mathrm{JSD}}$ using input sequences sampled from WikiText-103 (Merity et al., 2016; Xu et al., 2024) (consistent with prior work), and we compute $\phi_{H^{(\ell)}}$ using inputs sequences of tokens sampled uniformly at random from the vocabulary. We report results for a subset of these pairs involving base model Llama-2-7b-hf in Table 1 while deferring the rest and the full experimental setup details to Appendix C.

| | | | p-values | | |
|---|---|---|---|---|---|
| $\theta_1 = $ Llama-2-7b-hf, $\theta_2 =$ | Independent? | $\phi_{\mathrm{JSD}}$ (log) | $\phi_{\ell_2}$ | $\phi_{U^{(\ell)}}$ | $\phi_{H^{(\ell)}}$ |
| llama-7b-hf | ✓ | -11.10 | 0.98 | 0.60 | 0.25 |
| vicuna-7b-v1.1 | ✓ | -10.40 | 0.63 | 0.16 | 0.64 |
| Amber | ✓ | -10.69 | 0.75 | 0.36 | 0.88 |
| open-llama-7b | ✓ | -8.38 | 0.26 | 0.36 | 0.71 |
| vicuna-7b-v1.5 | ✗ | -10.87 | 0.01 | $\varepsilon$ | $\varepsilon$ |
| CodeLlama-7b-hf | ✗ | -10.62 | 0.01 | $\varepsilon$ | $\varepsilon$ |
| llemma-7b | ✗ | -10.24 | 0.01 | $\varepsilon$ | $\varepsilon$ |
| Orca-2-7b | ✗ | -10.34 | 0.01 | $\varepsilon$ | $\varepsilon$ |

Table 1: Results of various test statistics with $\theta_1$ as Llama-2-7b-hf and $\theta_2$ ranging over the listed models. The "independent" column is the ground truth. Here, $\varepsilon = 2.2\mathrm{e}\text{-}308$ (numerical underflow for a 64-bit float).

Consistent with prior work Xu et al. (2024), we find that $\phi_{\mathrm{JSD}}$ does not reliably distinguish independent versus dependent model pairs. For example, CodeLlama-7b-hf exhibits a larger divergence with Llama-2-7b-hf than the independently-trained models llama-7b-hf and Amber.

All other test statistics reliably distinguish independent versus dependent pairs; in particular, the p-values we obtain using the other test statistics are negligible for all dependent pairs (for $\phi_{\ell_2}$, because we run PERMTEST with $T = 99$ for computational reasons, we cannot obtain a p-value less than 0.01.[3]). Notably, in contrast to our findings, prior work (Xu et al., 2024) argued that the $\ell_2$ distance between model parameters is not a reliable indicator of independence, in the sense that the $\ell_2$ distance between dependent pairs is sometimes larger than that of independent pairs (similar to the case of $\phi_{\mathrm{JSD}}$); the key difference is that Xu et al. (2024) report the raw $\ell_2$ distance whereas we obtain p-values from the raw distances using PERMTEST. We hypothesize that PERMTEST effectively standardizes the raw distances.

At the finetune level, consistent with our problem formulation, we treat any two finetunes of the same base model as dependent (i.e. the last scenario in Figure 1). Table 5 in Appendix C.1 reports p-values between vicuna-7b-v1.5 (a finetune of Llama-2-7b-hf) and other models, and consequently we find that the p-values for all tests are low between vicuna-7b-v1.5 and other finetunes of Llama-2-7b-hf, even though they are not direct fine-tunes of each other.

## 4.2 ADVERSARIAL SETTING

We evaluate the robust statistic $\phi_{\mathrm{MATCH}}$ on the same model pairs, using input sequences sampled from WikiText-103, except that we randomly permute and rotate the weight matrices of the second model (with the output preserved) to simulate an adversary, and report the results in Appendix C.2. We find that the distribution of $\phi_{\mathrm{MATCH}}$ on independent model pairs is close to uniform on $[0, 1]$ (Figure 3), whereas across all non-independent model pairs the statistic is at most $\varepsilon$. Unlike the non-adversarial setting, where the p-values are valid by construction, the output of the robust test does not enjoy such theoretical guarantees; however, Figure 3 suggests that even in the adversarial setting that we can treat $\phi_{\mathrm{MATCH}}$ as a p-value.

### 4.2.1 MLP RETRAINING

Since these robust tests rely on the intermediate MLP activations, an adversary could attempt to retrain only the MLP layers while fixing the rest of the model to fool these tests. We tried such MLP retraining by re-initializing the gate, up, and down projection matrices and feeding random inputs $x \in \mathbb{R}^{N \times n \times d_{\mathrm{embed}}}$ through the original and new MLP blocks and minimizing MSE loss of the outputs (retraining one MLP at a time). To ensure low

---

[3]Though computing $\phi_{\ell_2}^{(i)}$ over each MLP block and aggregating with FISHER yields p-values less than 1e-30 as well.

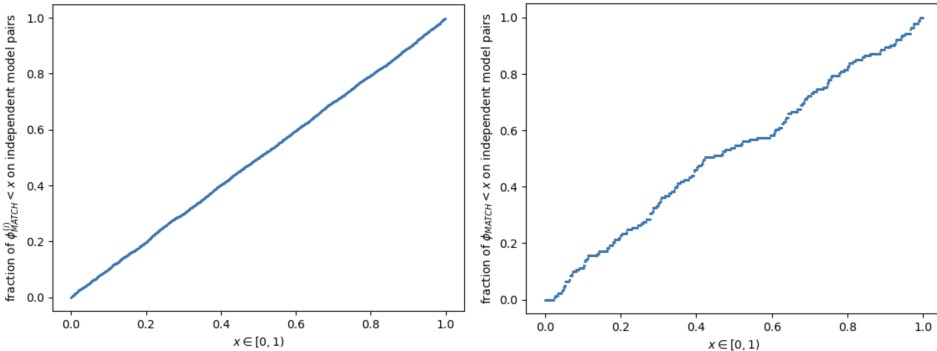

(a) Plot of $x \in [0,1)$ vs. the fraction of $\phi_{\text{MATCH}}^{(i)}$ (across all MLP blocks) of independent model pairs less than $x$.

(b) Plot of $x \in [0,1)$ vs. the fraction of $\phi_{\text{MATCH}}$ ($\phi_{\text{MATCH}}^{(i)}$ aggregated with `FISHER` across all MLP blocks) of independent model pairs less than $x$.

Figure 3: We find that $\phi_{\text{MATCH}}$ empirically acts as a p-value, as both plots roughly follow the line $y = x$.

| $\theta_1 = \texttt{Llama-2-70b-hf}, \theta_2 =$ | $\phi_{U^{(\ell)}}$ |
|---|---|
| `miqu-1-70b-pytorch` | $\varepsilon$ |
| `Llama-3.1-70B` | 0.571 |
| `Palmyra-Fin-70B-32K` | 0.539 |

Table 2: Results of $\phi_{U^{(\ell)}}$ (aggregated with `FISHER`) with $\theta_1$ as `meta-llama/Llama-2-70b-hf` and $\theta_2$ ranging over the listed models.

loss, we double the width of the MLP layers and compute the cosine similarity matrices and matchings the same way.

We individually retrain each of the 32 MLP layers (keeping other layers fixed) of `vicuna-7b-v1.5` (a fine-tune of `Llama-2-7b-hf`) for 10k gradient steps (until the loss curve plateaus). (Additional hyperparameters and a learning curve are in Appendix F.) For all 32 runs, we compare the robust statistic of the retrained model with original `Llama-2-7b-hf` and find our tests are robust to MLP retraining. For example, retraining the first MLP layer (with a final train loss of 0.0048), the value of the statistic $\phi_{\text{MATCH}}^{(1)}$ on the first MLP was less than $\varepsilon = 2.2\text{e-}308$, indicating that the two models are not independent. We find the same is true for the other MLP layers as well (i.e. $\phi_{\text{MATCH}}^{(\ell)}$ when evaluated on retrained layer $\ell$), with full results in Table 8 of Appendix F. Retraining the MLP layers could lead to a more expressive transformation of the model weights over simple permutations or rotations, and yet $\phi_{\text{MATCH}}^{(\ell)}$ remains small on the retrained non-independent models suggesting that the statistic is robust to retraining.

### 4.2.2 INDEPENDENT, IDENTICALLY DISTRIBUTED MODELS

We further evaluate the efficacy of our tests through ablations by training two near-identical models that only differ on select sources of randomness. We train two OLMo (7B) architecture models (Groeneveld et al. (2024)) on the same Dolmo dataset (Soldaini et al. (2024)), but with independently chosen initialization and data ordering, so we have two models that are essentially as similar as two independent models can be. We evaluate the statistics $\phi_{U^{(\ell)}}, \phi_{H^{(\ell)}}$, and $\phi_{\text{MATCH}}$ on the two models at four different training checkpoints, reported in Table 9 of Appendix G. We find that p-values for all statistics and checkpoints are broadly distributed, validating our tests can support independence even on two similarly-trained but independent models.

### 4.3 VARYING MODEL ARCHITECTURES

We evaluate our tests on models with different architectures—including between model pairs with different dimensions. First, we run $\phi_{U^{(\ell)}}$ on four 70B parameter models with the Llama 2-70B architecture shown in Table 2, and in particular, we verify that Miqu-70B is not independent from Llama 2-70B.

We also validate our tests on the Mistral architecture—we compare the weights of `StripedHyena-Nous-7B` (Poli et al. (2023)) with `Mistral-7B-v0.1` and find non-independent parameters via $\phi_{U^{(\ell)}}$. (We run $\phi_{U^{(\ell)}}$

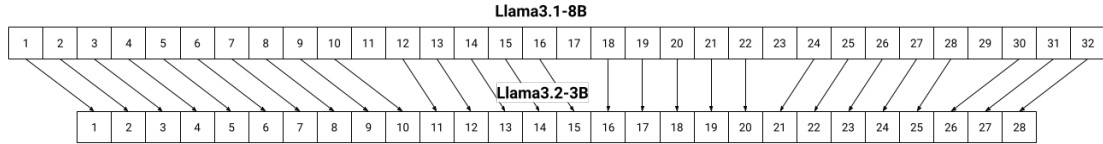

Figure 4: Matched Transformer blocks between Llama3.1-8B and Llama3.2-3B using $\phi_{\text{MATCH}}$, with arrows indicating if $\phi_{\text{MATCH}}^{(i,j)} <$ 1e-4.

on all parameters, not just the gate projection matrix of the MLPs; this allows us to identify similarity between specific layers.) Values of $\phi_{U^{(\ell)}}$ among parameters of the embedding layer and first Transformer block are shown in Table 10 in Appendix H.1. From the low p-values, some parameters, including the embedding layer and self-attention matrices of the first block were likely shared between the two models.

We also run the test on encoder-only models, e.g. BERT models, and on smaller models, like the 3B-parameter Phi-3 models. In these different architectures, the tests still provide strong signal for two non-independent models. Both $\phi_{H^{(\ell)}}$ and $\phi_{\text{MATCH}}$ return a statistic of $\varepsilon$ on the fine-tuned pair `Phi-3.5-mini-instruct` and `NuExtract-v1.5`. Similarly, $\phi_{H^{(\ell)}}$ on `bert-base-uncased` and finetune `tweets-gender-classifier-distilbert` yields a p-value less than $\varepsilon$ as well.

### 4.3.1 FINEGRAINED FORENSICS

The robust test can be run on a model pair of different architecture or dimension, as LAP can match a permutation for matrices $H_{\text{up}}(\theta_1)$ and $H_{\text{up}}(\theta_2)$ of different dimension (and returns a permutation of the smaller dimension). We run the robust test on all pairs of MLP blocks, i.e. computing

$$\phi_{\text{MATCH}}^{(i,j)} := \rho(\texttt{MATCH}(H_{\text{up}}^{(i)}(\theta_2), H_{\text{up}}^{(j)}(\theta_1)), \texttt{MATCH}(H_{\text{gate}}^{(i)}(\theta_2), H_{\text{gate}}^{(j)}(\theta_1))).$$

for all $i \in \{1, \ldots, L_1\}, j \in \{1, \ldots, L_2\}$ of models $\theta_1, \theta_2$ The flexibility of $\phi_{\text{MATCH}}$ for varying dimension, and running $\phi_{\text{MATCH}}$ on all pairs of MLP blocks, is significant because it prevents adversaries that may take only certain layers, or even only certain activations, of a pre-trained model and inject other layers.

In particular, we were able to identify the specific Transformer blocks of `Llama-3.1-8B` whose weights were likely used in initializing `Llama-3.2-3B` and `Llama-3.2-1B`, as Meta reported that the first two models were pruned from the third (MetaAI (2024)). We use $\phi_{\text{MATCH}}$ on all pairs of MLP blocks, and report the matched layers (identifying when $\phi_{\text{MATCH}}^{(i,j)}$ less than 1e-4) in Figure 4 and in Appendix H.2.

By comparing the aligned permutation returned from $\phi_{\text{MATCH}}$, we can even identify which hidden units were most likely shared between the blocks when MLP dimension is reduced from 14336 to 8192, for example (during the pruning process), shown in Appendix H.2.1. We also run this pairwise layer matching on the ShearedLlama (Xia et al. (2024)) models, which were the Llama 2-7B models pruned down to 1.3B and 2.7B parameters and find matching blocks, as well as on the pruned `nvidia/Llama-3.1-Minitron-4B-Depth-Base` model (Muralidharan et al. (2024)) from Llama 3.1, which we report in Appendix H.2.

## 5 CONCLUSION AND FUTURE WORK

In this paper, we propose and analyze various methods for detecting model independence in open-weight language models. Our robust method accurately predicts model relationships without training-time interventions while allowing some robustness to adversarial attacks with strong results within the Llama model family. Our methods demonstrate resilience to certain adversarial techniques, and we also investigate these methods across diverse model architectures. We demonstrate how our techniques provide fine-grain information about how one model may be derived from another and how they are robust to adversarial attacks. Yet, we do not exhaustively disprove the existence of a successful adversarial attack, and note that our tests are susceptible to false negatives. There is also an open question as to whether reliably differentiating between fine-tunes of the same base model to reconstruct a complete "family tree" of model lineage is possible (e.g. infer Llemma is a direct finetune of CodeLlama) (Yax et al., 2024). Furthermore, the fundamental question of whether robustness against adversarial attacks is solvable with exact guarantees warrants further exploration.

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

## A  TRANSFORMER ARCHITECTURE AND NOTATION

We consider models with the Llama Transformers architecture and define the notation henceforth, although this can easily be extended to other Transformer architectures.

Following the definition of $f_{\text{mlp}}$ in 3, we can define an abstraction of the full Llama language model architecture consisting of $L$ Transformer blocks sandwiched between an input and output layer. For the sequel, we will abuse notation in applying $f_{\text{mlp}}$ to multi-dimensional tensors by broadcasting along the last axis. We use $d, n \in \mathbb{N}$ to respectively denote the model dimension and sequence length, where $\Theta_{\text{LM}} = \Theta_{\text{in}} \times \Theta_{\text{block}}^{\times L} \times \Theta_{\text{out}}$ with $\Theta_{\text{block}}$ denoting the parameter space of each Transformer block and $\Theta_{\text{in}}, \Theta_{\text{out}}$ denoting the parameter spaces the input and output layers. We decompose $\Theta_{\text{block}} = \Theta_{\text{attn}} \times \Theta_{\text{mlp}}$ and use $f_{\text{rest}} : \Theta_{\text{attn}} \times \mathbb{R}^{n \times d} \to \mathbb{R}^{n \times d}$ to denote all remaining parts of the Transformer besides the MLP. The inputs to $f_{\text{rest}}$ are the input and output of the MLP, and the output of $f_{\text{rest}}$ is fed directly to the MLP of the next layer. In particular, $f_{\text{rest}}$ takes the input and output to the MLP of layer $i$, and first performs the residual connection following the MLP of layer $i$, then the self-attention and normalization components of layer $i + 1$, and returns the input to the MLP of layer $i + 1$. We use

| Parameter name | Notation |
|---|---|
| embedding | $E \in \mathbb{R}^{V \times d_{\text{emb}}}$ |
| input layernorm | $\gamma_{\text{input},i} \in \mathbb{R}^{1 \times d_{\text{emb}}}$ |
| attention query matrix | $W_{Q,i} \in \mathbb{R}^{d_{\text{emb}} \times d_{\text{emb}}}$ |
| attention key matrix | $W_{K,i} \in \mathbb{R}^{d_{\text{emb}} \times d_{\text{emb}}}$ |
| attention value matrix | $W_{V,i} \in \mathbb{R}^{d_{\text{emb}} \times d_{\text{emb}}}$ |
| attention output matrix | $W_{O,i} \in \mathbb{R}^{d_{\text{emb}} \times d_{\text{emb}}}$ |
| post-attention layernorm | $\gamma_{\text{post-attn},\, i} \in \mathbb{R}^{1 \times d_{\text{emb}}}$ |
| MLP gate projection | $G_i \in \mathbb{R}^{d_{\text{mlp}} \times d_{\text{emb}}}$ |
| MLP up projection | $U_i \in \mathbb{R}^{d_{\text{mlp}} \times d_{\text{emb}}}$ |
| MLP down projection | $D_i \in \mathbb{R}^{d_{\text{emb}} \times d_{\text{mlp}}}$ |
| final layernorm | $\gamma_{\text{final}} \in \mathbb{R}^{1 \times d_{\text{emb}}}$ |
| linear output | $O \in \mathbb{R}^{d_{\text{emb}} \times V}$ |

Table 3: Llama model architecture and dimensions.

$f_{\text{in}} : \Theta_{\text{in}} \times \mathcal{X} \to \mathbb{R}^{n \times d}$ and $f_{\text{out}} : \Theta_{\text{block}}^{(L)} \times \mathbb{R}^{n \times d} \to \mathcal{Y}$ to respectively denote the input and output layers, i.e. the elements before the first MLP and after the last MLP. Putting everything together gives the following definition of the model; we introduce the notation $X_\theta^{(i)}$ in the definition as a matter of convenience to track intermediate activations.

**Definition 4.** (GLU Transformer model) Let $\theta = (\theta_{\text{in}}, \{\theta_{\text{block}}^{(i)}\}_{i=1}^L, \theta_{\text{out}}) \in \Theta_{\text{LM}}$ and $X \in \mathcal{X}$, with $\theta_{\text{block}}^{(i)} = (\theta_{\text{attn}}^{(i)}, \theta_{\text{mlp}}^{(i)})$. Then $f_{\text{LM}}(X; \theta) = f_{\text{out}}(X_\theta^{(L)}; \theta_{\text{out}})$ for $X_\theta^{(0)} = f_{\text{in}}(X; \theta_{\text{in}})$ and

$$X_\theta^{(i)} = f_{\text{rest}}(X_\theta^{(i-1)}, f_{\text{mlp}}(X_\theta^{(i-1)})). \tag{3}$$

For a Llama model, table 3 describes the shapes of the model weight matrices for $i = 1, \dots, L$, for $V$ (vocab size), $d_{\text{emb}}$ (the hidden dimension), and $d_{\text{mlp}}$ (MLP hidden dimension). Following Definition 4, we have $\theta_{\text{in}} = (E), \theta_{\text{block}}^{(i)} = (\theta_{\text{attn}}^{(i)}, \theta_{\text{mlp}}^{(i)})$ where $\theta_{\text{attn}}^{(i)} = (\gamma_{\text{input},i}, W_{Q,i}, W_{K,i}, W_{V,i}, W_{O,i}, \gamma_{\text{post-attn}}^{(i)}), \theta_{\text{mlp}}^{(i)} = (G_i, U_i, D_i)$, and $\theta_{\text{out}} = (\gamma_{\text{final}}, L)$. We now describe a forward pass of the model.

We define the softmax function on a vector $v = (v_1, \dots, v_n)$, softmax$(v)$, as

$$\text{softmax}(v)_i = \frac{e^{v_i}}{\sum_{k=1}^n e^{v_k}}.$$

On batched input $X \in \mathbb{R}^{N \times n \times m}$ where each $X^{(b)} = [w_1 | \dots | w_m] \in \mathbb{R}^{n \times m}$ with column vectors $w_i$, we define the softmax as

$$\text{softmax}(X^{(b)}) = [\text{softmax}(w_1) | \dots | \text{softmax}(w_m)],$$

$$\text{softmax}(X) = [\text{softmax}(X^{(1)}) | \dots | \text{softmax}(X^{(N)})].$$

For a forward pass of the model $f_{\text{LM}}(X; \theta)$, consider an input sequence of tokens $X \in \{0, 1\}^{N \times V}$ as one-hot vectors where $n$ is sequence length. Then

We feed the input through:

1. ($f_{\text{in}}$) Embedding layer:
$$X_\theta^{(0)} = f_{\text{in}}(X; \theta_{\text{in}}) = XE \in \mathbb{R}^{N \times d_{\text{emb}}}$$

2. ($f_{\text{attn}}, f_{\text{mlp}}, f_{\text{post}}$) For each Transformer block $i = 0, 1, \dots, L$, through $f_{\text{attn}}, f_{\text{mlp}}$, and $f_{\text{post}}$:

   (a) Input layernorm:
   $$X_{\text{LN}_1}^{(i)} = \frac{X_\theta^{(i)}}{\sqrt{\text{Var}(X_\theta^{(i)}) + \varepsilon}} \odot \gamma_{\text{input},i}$$

   (with variance over the last axis) for some offset $\varepsilon$ (typically 1e-6).

(b) Causal multi-head self-attention: Split $X_{\mathrm{LN}_1}^{(i)}$ on the first axis into nheads $X_{\mathrm{LN}_1,j}^{(i)}, \ldots, X_{\mathrm{LN}_1,\mathrm{nheads}}^{(i)}$.
On each head $X_{\mathrm{LN}_1,j}^{(i)}$,

$$X_{\mathrm{SA},j}^{(i)} = \text{self-attn}(X_{\mathrm{LN}_1,j}^{(i)}) = \text{softmax}\left(\frac{X_{\mathrm{LN}_1,j}^{(i)} W_{Q,i}^T (X_{\mathrm{LN}_1,j}^{(i)} W_{K,i}^T)^T}{\sqrt{d_{\mathrm{emb}}}}\right) X_{\mathrm{LN}_1,j}^{(i)} W_{V,i}^T W_{O,i}^T$$

and concatenate $X_{\mathrm{SA},j}^{(i)}$ along the first axis again as $X_{\mathrm{SA}}^{(i)}$.

(c) Dropout and residual connection: $X_{\mathrm{DR}_1}^{(i)} = X_{\mathrm{LN}_1}^{(i)} + \text{Dropout}(X_{\mathrm{SA}}^{(i)})$

(d) Post-attention layernorm:

$$X_{\mathrm{LN}_2}^{(i)} = \frac{X_{\mathrm{DR}_1}^{(i)}}{\sqrt{\text{Var}(X_{\mathrm{DR}_1}^{(i)}) + \varepsilon}} \odot \gamma_{\text{post-attn},i}$$

(with variance over the last axis) for some offset $\varepsilon$. Then we have

$$f_{\text{attn}}(X_\theta^{(i-1)}; \theta_{\text{attn}}^{(i)}) = X_{\mathrm{LN}_2}^{(i)}.$$

(e) Next, we feed through $f_{\text{mlp}}$, the multi-layer perceptron:

$$f_{\text{mlp}}(X_{\mathrm{LN}_2}^{(i)}; \theta_{\text{mlp}}^{(i)}) = X_i^{\text{MLP}} = [\sigma(X_i^{\mathrm{LN}_2} G_i^T) \odot (X_i^{\mathrm{LN}_2} U_i^T)]D_i^T$$

for some activation $\sigma$ (e.g., SiLU).

(f) Finally, we feed through $f_{\text{post}}$, dropout and the residual connection:

$$f_{\text{post}}(\theta_{\text{mlp}}^{(i)}) = X_\theta^{(i+1)} = X_i^{\mathrm{DR}_1} + \text{Dropout}(X_i^{\text{MLP}})$$

3. ($f_{\text{out}}$) Final layernorm on the output $X_\theta^{(N+1)}$ from the final Transformer block:

$$X_{\mathrm{LN}}^{(L)} = \frac{X_\theta^{(L)}}{\sqrt{\text{Var}(X_\theta^{(L)}) + \varepsilon}} \odot \gamma_{\text{final}}$$

(with variance over the last axis) for some offset $\varepsilon$. Then, linear output embedding and softmax mapping to output probabilities:

$$f_{\text{out}}(X_\theta^{(L)}) = \text{softmax}(X_{\mathrm{LN}}^{(L)} O^T),$$

which defines the entire forward pass $f_{\mathrm{LM}}(X; \theta)$.

## B  MODEL PERMUTATION

We describe two sets of equivariant transformations $\Pi$ on a Transformer model as described in Appendix A. (Abusing notation), the first set, $\Pi_{\text{emb}}$, consists of elements $\pi_{\text{emb}}$ where $\pi_{\text{emb}} \in \mathbb{R}^{d_{\text{emb}} \times d_{\text{emb}}}$ is a permutation matrix. The second set, $\Pi_{\text{mlp}}$, consists of elements $\pi_{\text{mlp}}$ where $\pi_{\text{mlp}} \in \mathbb{R}^{d_{\text{mlp}} \times d_{\text{mlp}}}$ is a permutation matrix.

1. $\pi_{\text{emb}}(\theta)$: Applying an embedding permutation $\pi_{\text{emb}} \in \mathbb{R}^{d_{\text{emb}} \times d_{\text{emb}}}$ by left or right multiplying all relevant matrices by $\xi_{\text{embed}}$ (permuting rows or columns).

2. $\pi_{\text{mlp}}(\theta)$: Applying MLP permutations $\pi_{\text{mlp},i} \in \mathbb{R}^{d_{\text{mlp}} \times d_{\text{mlp}}}$ to MLP layers.

These permutations are applied such that the outputs of the original model $\theta$ and the permuted model $\Pi(\theta)$ remain aligned. We describe the details in Table 4.

## C  ADDITIONAL EXPERIMENTAL RESULTS ON LLAMA MODELS

### C.1  NON-ADVERSARIAL SETTING

First, we report statistics on various model pairs involving the base model `Llama-2-7b-hf` in Table 1 and various pairs involving `vicuna-7b-v1.5` in Table 5. Note `vicuna-7b-v1.5` is itself a finetune

| Parameter name | $\theta$ | $\pi_{\text{emb}}(\theta)$ | $\pi_{\text{mlp}}(\theta)$ |
|---|---|---|---|
| embedding | $E$ | $E\pi_{\text{emb}}$ | $E$ |
| input layernorm | $\gamma_{\text{input},i}$ | $\gamma_{\text{input},i}\pi_{\text{emb}}$ | $\gamma_{\text{input},i}$ |
| attention query matrix | $W_{Q,i}$ | $W_{Q,i}\pi_{\text{emb}}$ | $W_{Q,i}$ |
| attention key matrix | $W_{K,i}$ | $W_{K,i}\pi_{\text{emb}}$ | $W_{K,i}$ |
| attention value matrix | $W_{V,i}$ | $W_{V,i}\pi_{\text{emb}}$ | $W_{V,i}$ |
| attention output matrix | $W_{O,i}$ | $\pi_{\text{emb}}^T W_{O,i}$ | $W_{O,i}$ |
| post-attention layernorm | $\gamma_{\text{post-attn},\,i}$ | $\gamma_{\text{post-attn},\,i}\pi_{\text{emb}}$ | $\gamma_{\text{post-attn},\,i}$ |
| MLP gate projection | $G_i$ | $G_i\pi_{\text{emb}}$ | $\pi_{\text{mlp},i}G_i$ |
| MLP up projection | $U_i$ | $U_i\pi_{\text{emb}}$ | $\pi_{\text{mlp},i}U_i$ |
| MLP down projection | $D_i$ | $\pi_{\text{emb}}^T D_i$ | $D_i\pi_{\text{mlp},i}^T$ |
| final layernorm | $\gamma_{\text{final}}$ | $\gamma_{\text{final}}\pi_{\text{emb}}$ | $\gamma_{\text{final}}$ |
| linear output | $O$ | $\pi_{\text{emb}}^T O$ | $O$ |

Table 4: Transformations $\pi_{\text{emb}}$ and $\pi_{\text{mlp}}$ applied to a Llama-architecture model.

| | | | p-values | | |
|---|---|---|---|---|---|
| $\theta_1 = $ `vicuna-7b-v1.5`, $\theta_2 = $ | Independent? | $\phi_{\text{JSD}}$ (log) | $\phi_{\ell_2}$ | $\phi_{U^{(\ell)}}$ | $\phi_{H^{(\ell)}}$ |
| `llama-7b-hf` | ✓ | -10.39 | 0.40 | 0.29 | 0.59 |
| `vicuna-7b-v1.1` | ✓ | -10.41 | 0.63 | 0.12 | 0.29 |
| `Amber` | ✓ | -10.17 | 0.75 | 0.18 | 0.31 |
| `Llama-2-7b-hf` | ✗ | -10.87 | 0.01 | $\varepsilon$ | $\varepsilon$ |
| `CodeLlama-7b-hf` | ✗ | -10.10 | 0.01 | $\varepsilon$ | $\varepsilon$ |
| `llemma-7b` | ✗ | -9.87 | 0.01 | $\varepsilon$ | $\varepsilon$ |

Table 5: Results of various test statistics with $\theta_1$ as `lmsys/vicuna-7b-v1.5` and $\theta_2$ ranging over the listed models. Once again, $\varepsilon = 2.2e\text{-}308$.

of `Llama-2-7b-hf`. Consistent with our problem formulation (Section 3.1), we treat any finetune of `Llama-2-7b-hf` as dependent with `vicuna-7b-v1.5`, even in cases where neither model is a finetune of the other (i.e., the last scenario in

Next, we report p-values from the statistics $\phi_{\ell_2}, \phi_{U^{(\ell)}}$, and $\phi_{H^{(\ell)}}$ on all 210 model pairs (from 21 Llama 2-architecture models) in Figures 5, 6, and 7, where the model names are colored by base model (ground truth). For all statistics, the p-values on independent model pairs are uniformly distributed, while they are all significant at 0.01 (smaller for $\phi_{U^{(\ell)}}$ and $\phi_{H^{(\ell)}}$) for fine-tuned model pairs.

### C.2 ADVERSARIAL SETTING

We report values of $\phi_{\text{MATCH}}$ on all model pairs in Figure 8. The statistic is low ($< \varepsilon = 10^{-308}$) for all non-independent model pairs, and uniformly distributed for independent model pairs, empirically acting as a p-value.

## D ROBUST PROBLEM FORMULATION ADDENDUM

An adversary could apply a particular rotation scheme by multiplying weight matrices by an orthogonal rotation matrix $U$ that will also preserve outputs. We describe such a transformation which breaks the invariants proposed by Zeng et al. (2024) by manipulating layernorms. While this list may not be exhaustive, the following six transformations (with the first two described previously) "camouflage" the language model while preserving outputs:

T1. Permuting the rows of the embedding matrix (and subsequent matrices due to residual connections) by a permutation $\xi_{\text{emb}} \in \mathbb{R}^{d_{\text{emb}} \times d_{\text{emb}}}$

T2. Permuting the MLP matrices ($N$ different permutations for each Transformer block) by permutations $\xi_1, \ldots, \xi_N \in \mathbb{R}^{d_{\text{mlp}} \times d_{\text{mlp}}}$

T3. Rotating the embedding matrix (and subsequent matrices due to residual connections) by an orthogonal rotation matrix $R_{\text{emb}} \in \mathbb{R}^{d_{\text{emb}} \times d_{\text{emb}}}$

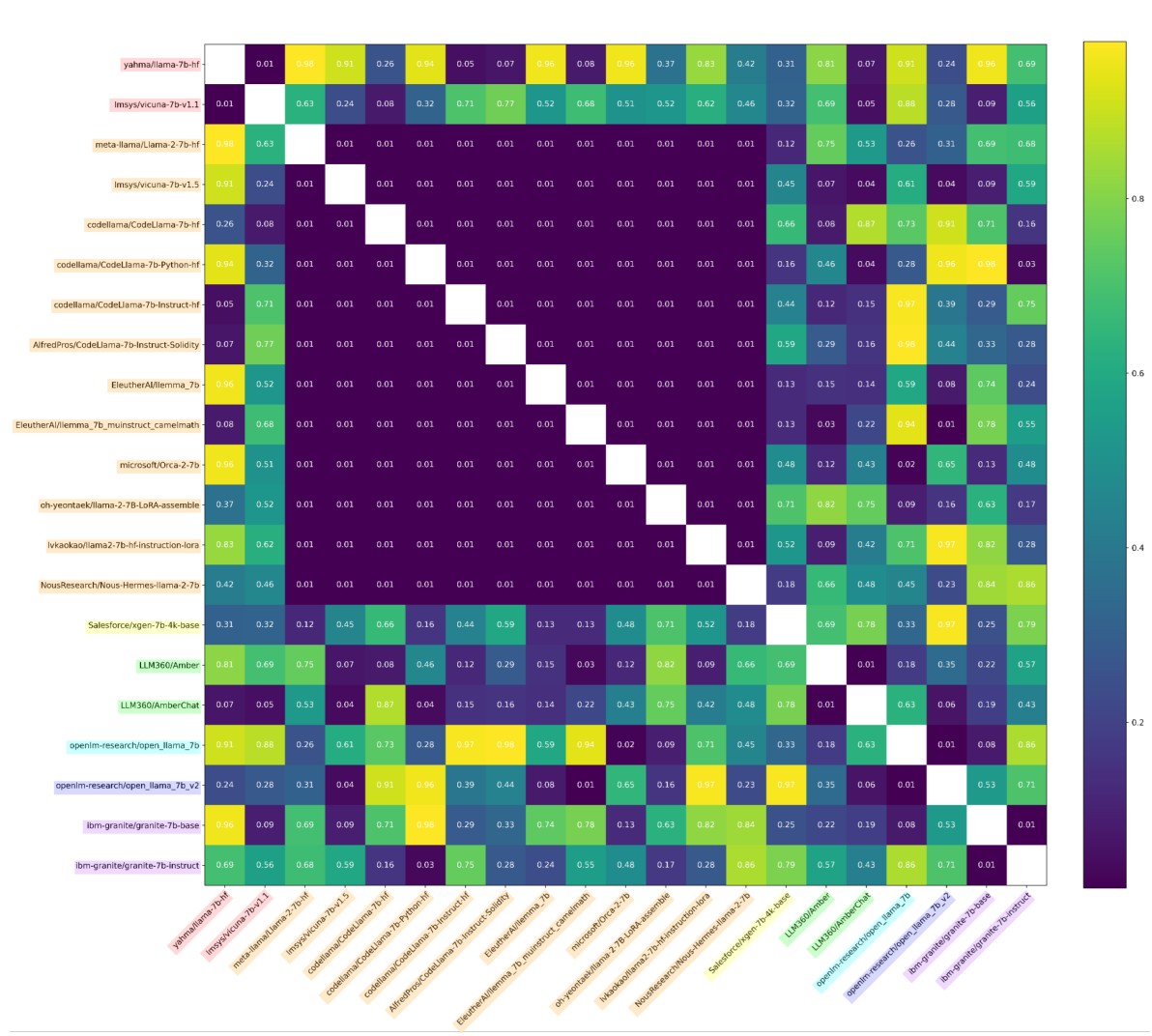

Figure 5: Results of p-values from $\phi_{\ell_2}$ on all model pairs.

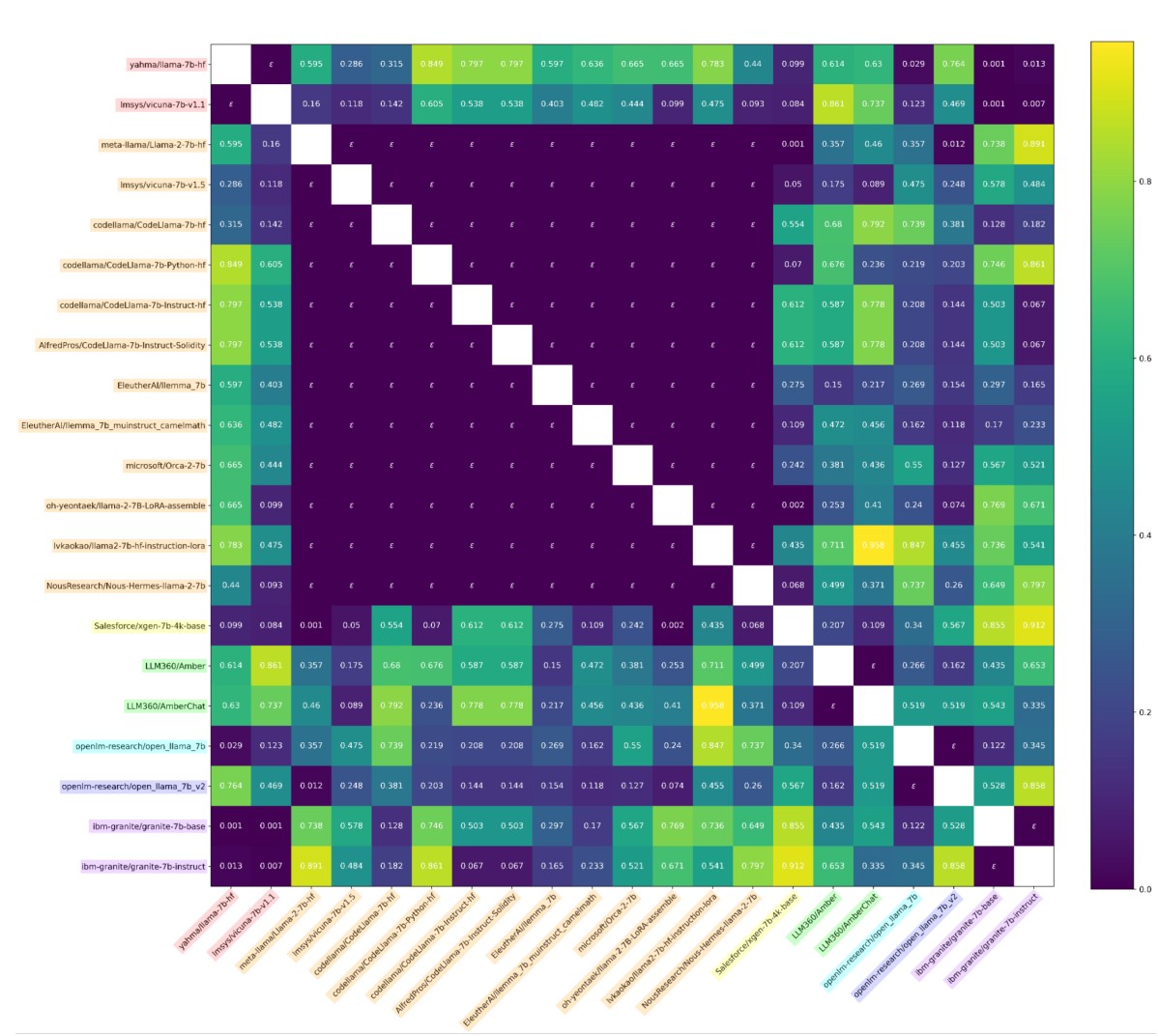

Figure 6: Results of p-values from $\phi_{U(\ell)}$ on all model pairs, where $\varepsilon = 2.2\text{e-}308$.

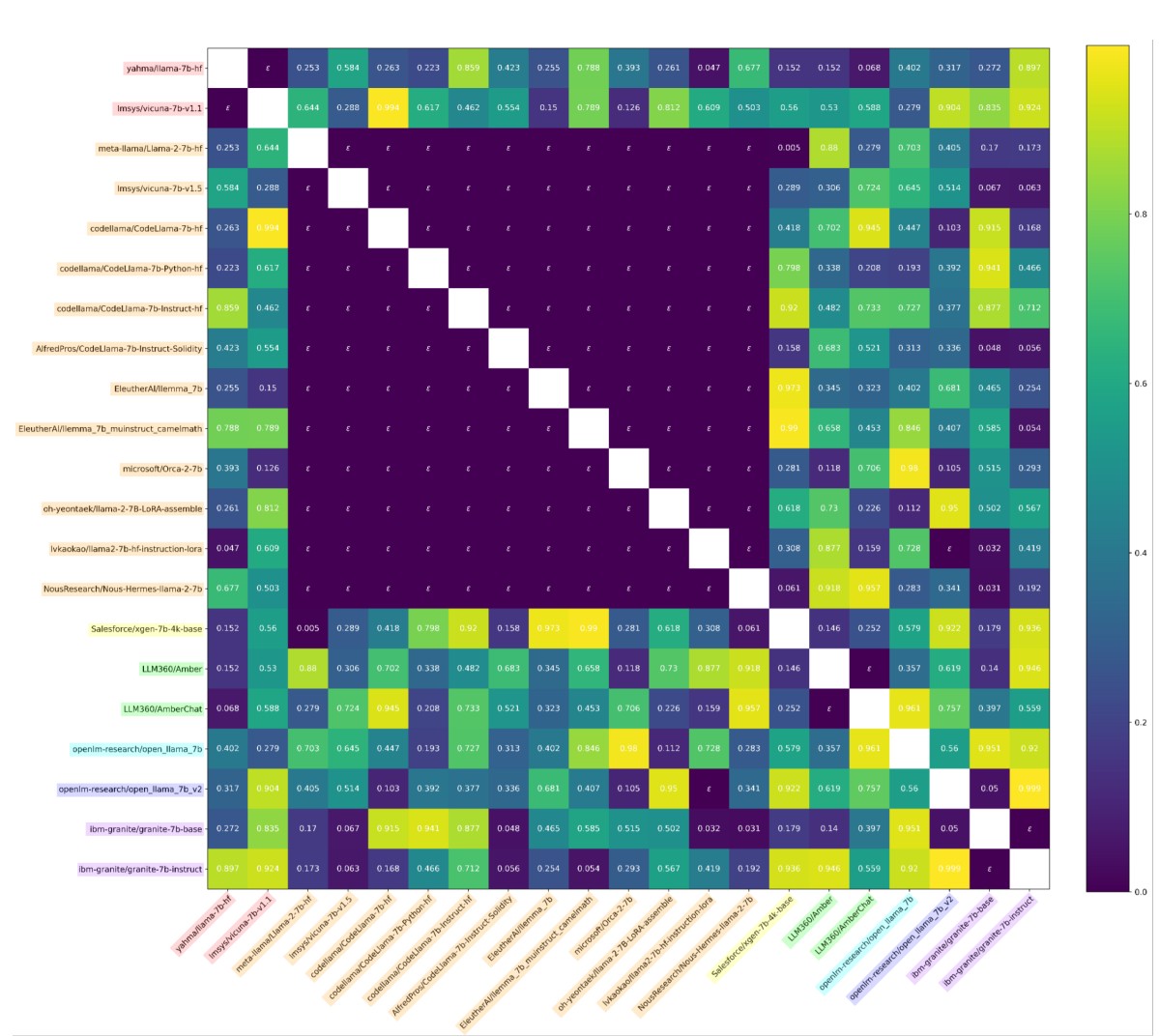

Figure 7: Results of p-values from $\phi_{H^{(\ell)}}$ on all model pairs, where $\varepsilon = 2.2\text{e-}308$.

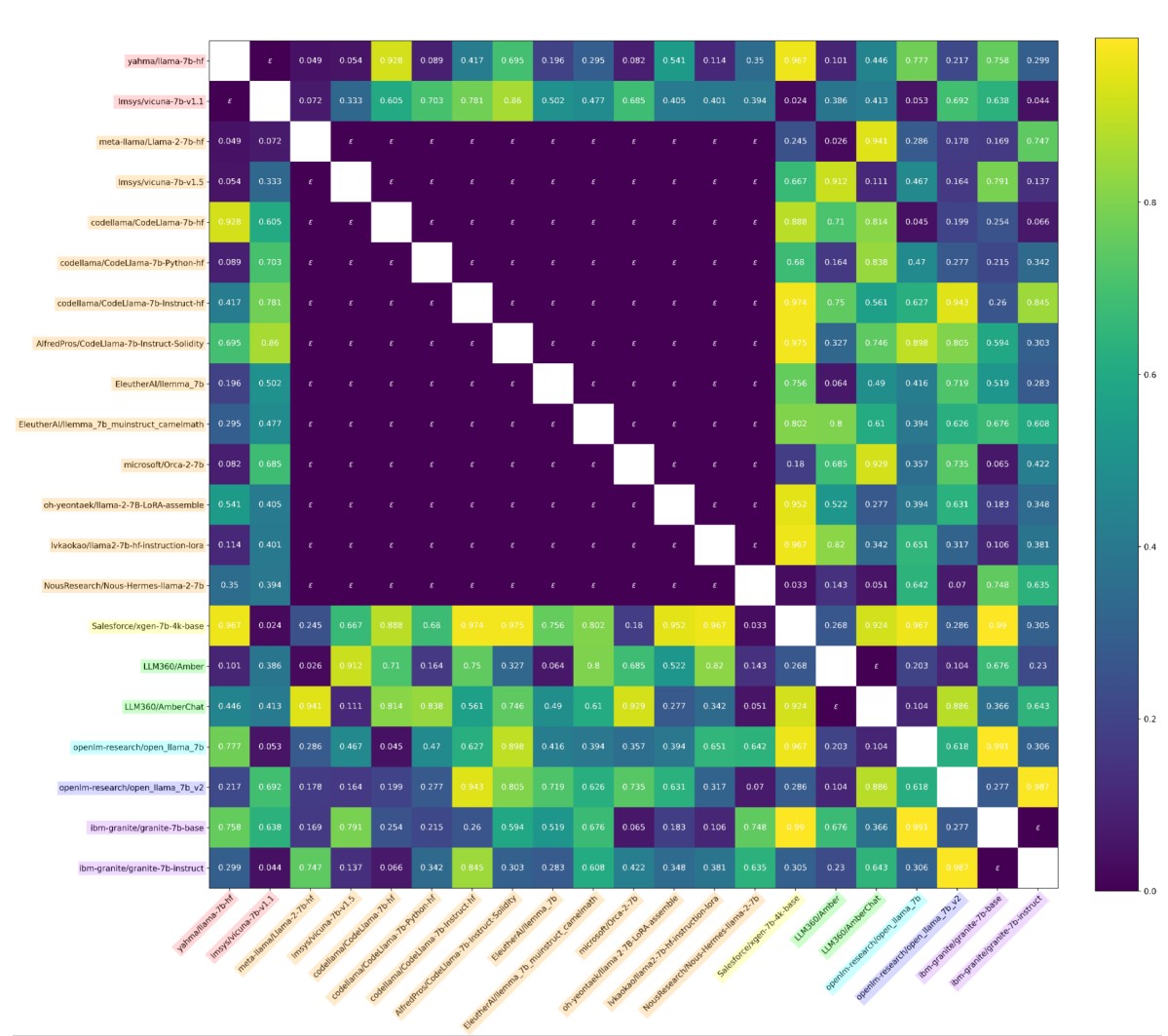

Figure 8: Results of values of $\phi_{\text{MATCH}}$ on all model pairs, where $\varepsilon = 2.2e\text{-}308$.

| **Parameter name** | $\theta$ | $\mathrm{Rot}(\theta) = \theta'$ |
|---|---|---|
| embedding | $E$ | $E R_{\mathrm{emb}}$ |
| input layernorm | $\gamma_{\mathrm{input},\,i}$ | $\gamma'_{\mathrm{input},\,i}$ |
| attention query matrix | $W_{Q,i}$ | $R_i\, W_{Q,i}\, \mathrm{diag}(\gamma_{\mathrm{input},\,i})\, R_{\mathrm{emb}}\, \mathrm{diag}(\frac{1}{\gamma'_{\mathrm{input},\,i}})$ |
| attention key matrix | $W_{K,i}$ | $R_i\, W_{K,i}\, \mathrm{diag}(\gamma_{\mathrm{input},\,i})\, R_{\mathrm{emb}}\, \mathrm{diag}(\frac{1}{\gamma'_{\mathrm{input},\,i}})$ |
| attention value matrix | $W_{V,i}$ | $W_{V,i}\, \mathrm{diag}(\gamma_{\mathrm{input},\,i})\, R_{\mathrm{emb}}\, \mathrm{diag}(\frac{1}{\gamma'_{\mathrm{input},\,i}})$ |
| attention output matrix | $W_{O,i}$ | $R_{\mathrm{emb}}^T\, W_{O,i}$ |
| post-attention layernorm | $\gamma_{\mathrm{post\text{-}attn},\,i}$ | $\gamma'_{\mathrm{post\text{-}attn},\,i}$ |
| MLP gate projection | $G_i$ | $G_i\, \mathrm{diag}(\gamma_{\mathrm{post\text{-}attn},i})\, R_{\mathrm{emb}}\, \mathrm{diag}(\frac{1}{\gamma'_{\mathrm{post\text{-}attn},i}})$ |
| MLP up projection | $U_i$ | $c_i U_i\, \mathrm{diag}(\gamma_{\mathrm{post\text{-}attn},i})\, R_{\mathrm{emb}}\, \mathrm{diag}(\frac{1}{\gamma'_{\mathrm{post\text{-}attn},i}})$ |
| MLP down projection | $D_i$ | $\frac{1}{c_i}\, R_{\mathrm{emb}}^T\, D_i$ |
| final layernorm | $\gamma_{\mathrm{final}}$ | $\gamma'_{\mathrm{final}}$ |
| linear output | $O$ | $O\, \mathrm{diag}(\gamma_{\mathrm{final}})\, R_{\mathrm{emb}}\, \mathrm{diag}(\frac{1}{\gamma'_{\mathrm{final}}})$ |

Table 6: Output-preserving rotation applied to a Llama-architecture model.

T4. Rotating the query and key attention matrices ($N$ different rotations for each Transformer block) by orthogonal rotation matrices $R_1, \ldots, R_N \in \mathbb{R}^{d_{\mathrm{emb}} \times d_{\mathrm{emb}}}$

T5. Replacing all layernorms (input, post-attention, final) with vectors in $\mathbb{R}^{1 \times d_{\mathrm{emb}}}$ with non-zero elements

T6. Scaling the MLP matrices by a constant non-zero factor

Consider a model $\theta$ of Llama architecture (Appendix A). Consider orthogonal matrices $R_{\mathrm{emb}}, R_1, \ldots R_{32}$ as described, as well as new layernorms $\gamma'_{\mathrm{input},1}, \ldots, \gamma'_{\mathrm{input},32}, \gamma'_{\mathrm{post\text{-}attn},1}, \ldots, \gamma'_{\mathrm{post\text{-}attn},32}$ in $\mathbb{R}^{1 \times d_{\mathrm{emb}}}$ with non-zero elements. Finally, consider non-zero constants $c_1, \ldots, c_{32}$, which we use to transform the layernorms. We apply the rotation with these parameters to $\theta$, to get a new "rotated" model, $\mathrm{Rot}(\theta)$. We generalize the set of transformations above as applying $\mathrm{Rot}(\theta)$ to a model $\theta$".

We transform all the original matrices of $\theta$ as in Table 6 (for $i = 1, \ldots, 32$). Note that the transformations T1 and T2 are elements of $\Pi_{\mathrm{emb}}$ and $\Pi_{\mathrm{mlp}}$ and the remaining transformations T3 to T6 are described in Table 6.

**Theorem 3.** *For any input sequence $X \in \{0,1\}^{n \times V}$, the outputs of models $\theta$ and $\mathrm{Rot}(\theta) = \theta'$ are aligned, i.e.* $f_{\mathrm{LM}}(X; \theta) = f_{\mathrm{LM}}(X; \theta')$.

*Proof.* First, note that an element-wise product of two one-dimensional vectors is equivalent to multiplying by the diagonal matrix of the second vector, i.e. for $v, \gamma \in R^{1 \times m}$,

$$v * \gamma = v \mathrm{diag}(\gamma).$$

We use this in our layernorm calculations.

Let the output from the unrotated embedding layer be $y = f_{\mathrm{in}}(X, E) = EX$ (for $X \in \{0,1\}^{n \times V}$). Then the output from the rotated embedding layer is $y' = f_{\mathrm{in}}(X, E') = (ER_{\mathrm{emb}})(x) = yR_{\mathrm{emb}}$. Now consider Transformer block $i$ with input $y$ and the rotated Transformer block with input $yR_{\mathrm{emb}}$. $y$ is passed into the input layernorm, which returns

$$z = LN_i(y) = \frac{y}{\sqrt{\mathsf{Var}(y) + \varepsilon}} \odot \gamma_{\mathrm{input},i} = \frac{y}{\sqrt{\mathsf{Var}(y) + \varepsilon}} \mathrm{diag}(\gamma_{\mathrm{input},i}).$$

The rotated input layernorm on $y'$ returns

$$z' = LN_i'(y') = \frac{y'}{\sqrt{\mathsf{Var}(y') + \varepsilon}} \odot \gamma'_{\mathrm{input},i} = \frac{yR_{\mathrm{emb}}}{\sqrt{\mathsf{Var}(yR_{\mathrm{emb}}) + \varepsilon}} \odot \gamma'_{\mathrm{input},i}$$

$$= \frac{y}{\sqrt{\mathsf{Var}(y) + \varepsilon}} R_{\mathrm{emb}} \mathrm{diag}(\gamma'_{\mathrm{input},i}) = z\, \mathrm{diag}(\frac{1}{\gamma_{\mathrm{input},i}}) R_{\mathrm{emb}} \mathrm{diag}(\gamma'_{\mathrm{input},i}),$$

which follows from $R_{\text{emb}}$ being orthogonal. Then we have the output from the unrotated self-attention is

$$w = \text{softmax}\left(\frac{zW_{Q,i}^T(zW_{K,i}^T)^T}{\sqrt{d_{\text{key}}}}\right)zW_{V,i}^T W_{O,i},$$

and the output from the rotated self-attention with input $z'$ is

$$\text{softmax}\left(\frac{z'(R_iW_{Q,i}\text{diag}(\gamma_{\text{input},\,i})R_{\text{emb}}\text{diag}(\frac{1}{\gamma'_{\text{input},\,i}}))^T(z'(R_iW_{K,i}\text{diag}(\gamma_{\text{input},\,i})R_{\text{emb}}\text{diag}(\frac{1}{\gamma'_{\text{input},\,i}}))^T)^T}{\sqrt{d_{\text{key}}}}\right)$$

$$z'(W_{V,i}\text{diag}(\gamma_{\text{input},\,i})R_{\text{emb}}\text{diag}(\frac{1}{\gamma'_{\text{input},\,i}}))^T(R_{\text{emb}}^T W_{O,i})^T$$

$$= \text{softmax}\left(\frac{z'\text{diag}(\frac{1}{\gamma'_{\text{input},\,i}})R_{\text{emb}}^T\text{diag}(\gamma_{\text{input},\,i})W_{Q,i}^T R_i^T(z'\text{diag}(\frac{1}{\gamma'_{\text{input},\,i}})R_{\text{emb}}^T\text{diag}(\gamma_{\text{input},\,i})W_{K,i}^T R_i^T)^T}{\sqrt{d_{\text{key}}}}\right)$$

$$z'\text{diag}(\frac{1}{\gamma'_{\text{input},\,i}})R_{\text{emb}}^T\text{diag}(\gamma_{\text{input},\,i})W_{V,i}^T W_{O,i}^T R_{\text{emb}}$$

$$= \text{softmax}\left(\frac{z'\text{diag}(\frac{1}{\gamma'_{\text{input},\,i}})R_{\text{emb}}^T\text{diag}(\gamma_{\text{input},\,i})W_{Q,i}^T W_{K,i}\text{diag}(\gamma_{\text{input},\,i})R_{\text{emb}}\text{diag}(\frac{1}{\gamma'_{\text{input},\,i}})(z')^T}{\sqrt{d_{\text{key}}}}\right)zW_{V,i}^T W_{O,i}^T R_{\text{emb}}$$

$$= \text{softmax}\left(\frac{zW_{Q,i}W_{K,i}^T z^T}{\sqrt{d_{\text{key}}}}\right)zW_{V,i}^T W_{O,i}^T R_{\text{emb}}$$

$$= wR_{\text{emb}} = w'.$$

Then $y$ and $y'$ respectively from before the layernorm are added as residual connections as $v = y + w$ and $v' = y' + w' = vR_{\text{emb}}$. $v$ is passed into the post-attention layernorm, which returns

$$u = LN_i(v) = \frac{v}{\sqrt{\text{Var}(v) + \varepsilon}} \odot \gamma_{\text{post-attn},i} = \frac{v}{\sqrt{\text{Var}(v) + \varepsilon}}\text{diag}(\gamma_{\text{post-attn},i}).$$

Similar to the input layernorm, the rotated post-attention layernorm on $v'$ returns

$$u' = LN_i'(v') = \frac{v'}{\sqrt{\text{Var}(v') + \varepsilon}} \odot \gamma'_{\text{post-attn},i} = \frac{vR_{\text{emb}}}{\sqrt{\text{Var}(vR_{\text{emb}}) + \varepsilon}} \odot \gamma'_{\text{post-attn},i}$$

$$= \frac{v}{\sqrt{\text{Var}(v) + \varepsilon}}R_{\text{emb}}\text{diag}(\gamma'_{\text{post-attn},i}) = u\,\text{diag}(\frac{1}{\gamma_{\text{post-attn},i}})R_{\text{emb}}\text{diag}(\gamma'_{\text{post-attn},i}).$$

Then the output from the unrotated MLP layer on $u$ is

$$t = [\sigma(uG_i^T) \odot (uU_i^T)]D_i^T$$

and the output from the rotated MLP on $u'$ is

$$t' = [\sigma(u'(G_i\text{diag}(\gamma_{\text{post-attn},i})R_{\text{emb}}\text{diag}(\frac{1}{\gamma'_{\text{post-attn},i}}))^T \odot (u'(c_iU_i\text{diag}(\gamma_{\text{post-attn},i})R_{\text{emb}}\text{diag}(\frac{1}{\gamma'_{\text{post-attn},i}}))^T)](\frac{1}{c_i}R_{\text{emb}}^T D_i)^T$$

$$= [\sigma(u\,\text{diag}(\frac{1}{\gamma_{\text{post-attn},i}})R_{\text{emb}}\text{diag}(\gamma'_{\text{post-attn},i})\text{diag}(\frac{1}{\gamma'_{\text{post-attn},i}})R_{\text{emb}}^T\text{diag}(\gamma_{\text{post-attn},i})G_i^T) \odot$$

$$(c_iu\,\text{diag}(\frac{1}{\gamma_{\text{post-attn},i}})R_{\text{emb}}\text{diag}(\gamma'_{\text{post-attn},i})\text{diag}(\frac{1}{\gamma'_{\text{post-attn},i}})R_{\text{emb}}^T\text{diag}(\gamma_{\text{post-attn},i}))U_i^T]\frac{1}{c_i}D_i^T R_{\text{emb}}$$

$$= [c_i\sigma(uG_i^T) \odot (uU_i^T)]\frac{1}{c_i}D_i^T R_{\text{emb}} = tR_{\text{emb}}.$$

Then the output from the self-attention is added as a residual connection, and the final output from the unrotated Transformer block is $s = t + v$, and the output from the rotated Transformer block is $s' = t' + v' = sR_{\text{emb}}$.

Suppose $a$ is the output after all Transformer layers in $\theta$ and $a'$ is the output after all Transformer layers in $\theta'$. Then the outputs after the final layernorms are

$$b = \frac{v}{\sqrt{\mathsf{Var}(a) + \varepsilon}} \mathrm{diag}(\gamma_{\mathrm{final}})$$

$$b' = b\, \mathrm{diag}(\frac{1}{\gamma_{\mathrm{final}}}) R_{\mathrm{emb}} \mathrm{diag}(\gamma'_{\mathrm{final}}),$$

and the logits from the linear output layer are

$$bO^T = b\, \mathrm{diag}(\frac{1}{\gamma_{\mathrm{final}}}) R_{\mathrm{emb}} \mathrm{diag}(\gamma'_{\mathrm{final}}) \mathrm{diag}(\gamma_{\mathrm{final}}) R_{\mathrm{emb}}^T \mathrm{diag}(\frac{1}{\gamma'_{\mathrm{final}}}) O^T$$
$$= b'(O')^T,$$

which are the same for both models. $\qquad\square$

We attempted to undo such a transformation that an adversary may apply by solving the least squares problem: We solve for a rotation $A$ that minimizes $|AX - Y|$ where $X$ is a weight matrix of the first model and $Y$ is the corresponding weight matrix of the second model. Although this will provide a potential rotation to undo this transformation, we find that this solution will also find a matrix $A$ that aligns two independent model pairs as well. This makes undo-ing the rotation this way unreliable. The same holds for $X$ and $Y$ that are activations over multiple inputs.

### D.1  HuREF Invariants

We also test and break the invariants from Zeng et al. (2024) with our transformation. We have that for rotated $M$, $M'$, and layer $i$, their first invariant is

$$M_a = E'(W'_{Q,i})^T((W'_{K,i})^T)^T E'^T$$

$$M'_a = (E R_{\mathrm{emb}}) \left( \mathrm{diag}(\frac{1}{\gamma'_{\mathrm{input},i}}) R_{\mathrm{emb}}^T \mathrm{diag}(\gamma_{\mathrm{input},i}) W_{Q,i}^T R_i^T \right) \left( R_i W_{K,i} \mathrm{diag}(\gamma_{\mathrm{input},i}) R_{\mathrm{emb}} \mathrm{diag}(\frac{1}{\gamma'_{\mathrm{input},i}}) \right) (R_{\mathrm{emb}}^T E)$$

$$= E R_{\mathrm{emb}} \mathrm{diag}(\frac{1}{\gamma'_{\mathrm{input},i}}) R_{\mathrm{emb}}^T \mathrm{diag}(\gamma_{\mathrm{input},i}) W_{Q,i}^T W_{K,i} \mathrm{diag}(\gamma_{\mathrm{input},i}) R_{\mathrm{emb}} \mathrm{diag}(\frac{1}{\gamma'_{\mathrm{input},i}}) R_{\mathrm{emb}}^T E,$$

and in general $M_a \neq M'_a$ unless the layernorm weights are equal constants. The other two invariants also do not hold due to changing the layernorms. (Note that our notation for Transformers is different than theirs.) Assuming in their invariant $M_f$ that $W_1$ and $W_2$ are the gate and down projection matrices of an MLP (this is not stated explicitly in the paper but can be inferred from experiments), the remaining invariants do not hold either.

Empirically, we compute the invariants between Llama2-7b and independently trained models and between Llama2-7b and rotated finetuned models (including Llama2-7b) in Table 7. We can see there is little distinction between the independent vs. non-independent model pairs.

| $\theta_1 = $ Llama-2-7b-hf, $\theta_2 = $ | Independent? | $M_a$ | $M_b$ | $M_c$ | $\phi_{\mathrm{MATCH}}$ | $\phi_{U^{(\ell)}}$ | $\phi_{H^{(\ell)}}$ | $\phi_{\mathrm{JSD}}$ |
|---|---|---|---|---|---|---|---|---|
| vicuna-7b-v1.5 | ✗ | 1.0 | 0.9883 | 0.9922 | $< \varepsilon$ | $< \varepsilon$ | $< \varepsilon$ | -10.874 |
| Nous-Hermes-llama-2-7b | ✗ | 1.0 | 1.0 | 1.0 | $< \varepsilon$ | $< \varepsilon$ | $< \varepsilon$ | -12.101 |
| llama-7b-hf | ✓ | 0.0884 | 0.0250 | 0.0400 | 0.049 | 0.595 | 0.253 | -11.102 |
| AmberChat | ✓ | 0.1289 | -0.0093 | 0.0198 | 0.941 | 0.460 | 0.279 | -10.281 |
| Openllama-v1 | ✓ | 0.1084 | 0.0076 | 0.0057 | 0.286 | 0.357 | 0.703 | -8.381 |
| Rotated Llama-2-7b-hf | ✗ | 0.0767 | 0.0908 | 0.1011 | $< \varepsilon$ | 0.517 | 0.323 | $-\infty$ |
| Rotated vicuna-7b-v1.5 | ✗ | 0.1553 | 0.0933 | 0.0977 | $< \varepsilon$ | 0.688 | 0.857 | -10.874 |
| Rotated Nous-Hermes-llama-2-7b | ✗ | 0.0332 | 0.0718 | 0.1060 | $< \varepsilon$ | 0.772 | 0.240 | -12.101 |

Table 7: Results for the three invariants $M_a, M_b, M_c$ from Zeng et al. (2024) between Llama-2-7b-hf and independent and non-independent models.

## E    ADDITIONAL ROBUST STATISTIC

We describe another statistic robust to the described transformations that does not yield results like p-values.

**(Median of max of cosine similarity of hidden activations:)** Consider two models $\theta_1$ and $\theta_2$ defined with parameters as in 3 and their $i$-th MLP blocks, $\theta_{1,\text{mlp}}^{(i)}$, $\theta_{2,\text{mlp}}^{(i)}$. First we undo the random MLP permutation, by feeding the same input token sequences $X$ through the gate projection matrices of layer $i$. We have the activation matrices (outputs from the gate projection operations) from the two models at layer $i$,

$$(H_{\theta_1,\text{gate}}^{(i)})_{:,j,k} = U_1^{(i)} f_{\text{attn}}(X_{\theta_1}^{(i-1)}; \theta_{\text{pre},1}^{(i)})_{j,k}$$

$$(H_{\theta_2,\text{gate}}^{(i)})_{:,j,k} = U_2^{(i)} f_{\text{attn}}(X_{\theta_2}^{(i-1)}; \theta_{\text{pre},2}^{(i)})_{j,k}$$

We find a permutation $\xi_{\text{match}}$ using `MATCH` that best aligns $H_{\theta_\ell,\text{gate}}^{(i)}$ and $H_{\theta_\ell,\text{gate}}^{(i)}$:

$$\xi_{\text{match}} = \texttt{MATCH}(H_{\theta_1,\text{gate}}^{(i)}, H_{\theta_2,\text{gate}}^{(i)})$$

and undo the MLP permutation (of the $i$-th MLP block) on $\theta_2$ by right-multiplying the $i$-th gate projection and up projection matrices $G_2^{(i)}$ and $U_2^{(i)}$ by $\xi_{\text{match}}$ and right-multiplying the down projection $D_2^{(i)}$ by $\xi_{\text{match}}^T$:

$$(G_j^{(i)})' = G_j^{(i)} \xi_{\text{match}}, (U_j^{(i)})' = U_j^{(i)} \xi_{\text{match}}, (D_j^{(i)})' = \xi_{\text{match}} D_j^{(i)}$$

for $j = 1, 2$. Next, we fix the $i$-th post-attention layernorms for both models to have weights of 1, by changing for both $M_1$ and $M_2$:

$$(G_j^{(i)})'' = (G_j^{(i)})' \text{diag}(\gamma_{\text{post-attn},i}), (U_j^{(i)})'' = (U_j^{(i)})' \text{diag}(\gamma_{\text{post-attn},i})$$

$$(\gamma_{\text{post-attn},j})'^{(i)} = 1 \in \mathbb{R}^{1 \times d_{\text{emb}}}$$

Next, consider $V \in \mathbb{R}^{n \times 1 \times d_{\text{mlp}}}$, $n$ random vectors of size $(1, d_{\text{mlp}})$ with values sampled from $\mathcal{N}(0,1)$. We use $V$ to sample "rotated" inputs to the MLP gate layers for models $\theta_1$ and $\theta_2$ as linear combinations of the rows:

$$X_1^{(i)} = V(G_1^{(i)})'', X_2^{(i)} = V(G_2^{(i)})''.$$

For this section, we assume matrix multiplications are batched. Then we have $X_1, X_2 \in R^{n \times 1 \times d_{\text{emb}}}$ and squeeze them to be in $R^{n \times d_{\text{emb}}}$. Now, $X_1$ and $X_2$ will match the rotations of their respective models, and we compute the cosine similarity of the activation matrices after feeding $X_1$ and $X_2$ through the gate projection layers of the $i$-th MLP blocks of $\theta_1$ and $\theta_2$, respectively:

$$A = \texttt{cossim}(X_1((G_1^{(i)})'')^T, X_2((G_2^{(i)})'')^T)$$

$$A_{jk} = \frac{(X_1((G_1^{(i)})'')^T)^{(j)} \cdot (X_2((G_2^{(i)})'')^T)^{(k)}}{\left\|(X_1((G_1^{(i)})'')^T)^{(j)}\right\| \left\|(X_2((G_2^{(i)})'')^T)^{(k)}\right\|}$$

(This will be invariant to an embedding permutation because the sampled inputs will match the embedding permutation applied to the MLP matrices.)

We find a threshold between the values from independent model pairs vs. non-independent model pairs for this statistic. For independent model pairs, we find the statistic is generally above 0.40 (often higher), and for fine-tuned pairs it is closer to 0.20. The histogram in Figure 9 shows the distribution of the statistic computed for the first MLP layer for independent (blue) vs. non-independent (green) model pairs.

## F    MLP RETRAINING EXPERIMENTS

We retrain each of the 32 MLP layers by feeding in random inputs through the original MLP (gate, up, and down projection matrices.) We train for 10000 gradient steps using MSE loss and an Adam Optimizer with a learning rate of 0.001 and batch size of 5000. A sample learning curve is in Figure 10.

The MLP retraining results for all 32 MLP layers of `vicuna-7b-v1.5`, compared with `Llama-2-7b-hf` are in Table 8, showing that the statistic is robust to retraining of all layers.

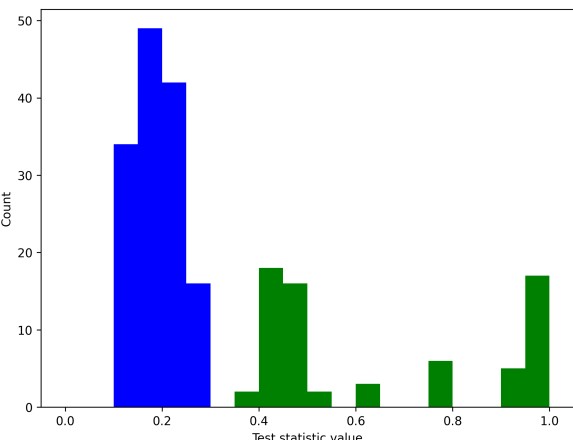

Figure 9: Histogram of median-max for model pairs

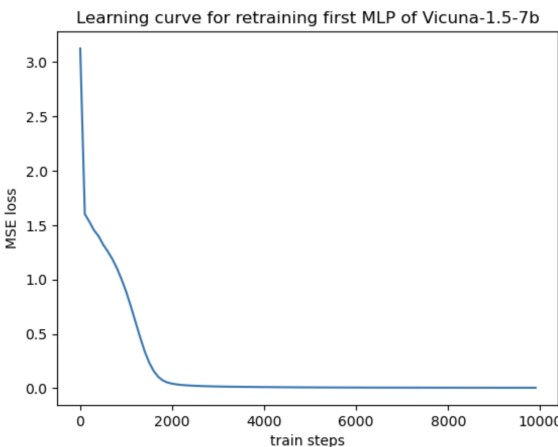

Figure 10: Learning curve for MLP retraining.

## G    INDEPENDENT, IDENTICALLY DISTRIBUTED MODELS

As described in Section 4.2.2, we ensure the validity of our tests on independently initialized, but very similar models. We randomly initialized a model with the OLMo (7B) architecture (Groeneveld et al., 2024) and

| MLP | Loss | $\log_{10}(\phi_{\text{MATCH}}^{(i)})$ | MLP | Loss | $\log_{10}(\phi_{\text{MATCH}}^{(i)})$ | MLP | Loss | $\log_{10}(\phi_{\text{MATCH}}^{(i)})$ |
|---|---|---|---|---|---|---|---|---|
| 1 | 0.0048 | $-479$ | 12 | 0.0060 | $-342$ | 23 | 0.0043 | $-593$ |
| 2 | 0.012 | $-485$ | 13 | 0.0058 | $-330$ | 24 | 0.0047 | $-542$ |
| 3 | 0.0026 | $-614$ | 14 | 0.0066 | $-323$ | 25 | 0.0050 | $-497$ |
| 4 | 0.0034 | $-580$ | 15 | 0.0063 | $-414$ | 26 | 0.0051 | $-534$ |
| 5 | 0.0030 | $-523$ | 16 | 0.0061 | $-394$ | 27 | 0.0052 | $-482$ |
| 6 | 0.0035 | $-513$ | 17 | 0.0063 | $-445$ | 28 | 0.0061 | $-477$ |
| 7 | 0.0041 | $-533$ | 18 | 0.0055 | $-515$ | 29 | 0.0065 | $-433$ |
| 8 | 0.0042 | $-464$ | 19 | 0.0045 | $-571$ | 30 | 0.0098 | $-361$ |
| 9 | 0.0050 | $-439$ | 20 | 0.0045 | $-512$ | 31 | 2.313 | $-26.4$ |
| 10 | 0.0050 | $-377$ | 21 | 0.0047 | $-595$ | 32 | 0.0114 | $-174$ |
| 11 | 0.0060 | $-365$ | 22 | 0.0043 | $-555$ | | | |

Table 8: $\phi_{\text{MATCH}}$ on individual blocks between `Llama-2-7b-hf` and `vicuna-7b-v1.5` after retraining MLP layers.

| # train tokens | $\phi_{U^{(\ell)}}$ | $\phi_{H^{(\ell)}}$ | $\phi_{\ell_2}$ | $\phi_{\text{MATCH}}$ | $\phi_{\text{JSD}}$ (log) |
|---|---|---|---|---|---|
| 100M | 0.641 | 0.119 | 0.07 | 0.809 | -11.81 |
| 1B | 0.789 | 0.483 | 0.06 | 0.443 | -11.05 |
| 10B | 0.707 | 0.277 | 0.93 | 0.343 | -11.28 |
| 18B | 0.819 | 0.141 | 0.64 | 0.027 | -11.03 |

Table 9: Results for $\phi_{U^{(\ell)}}, \phi_{H^{(\ell)}}$, and $\phi_{\text{MATCH}}$ evaluated on training checkpoints between two independently-trained OLMo models.

| Parameter name | Notation | $\phi_{U^{(\ell)}}$ |
|---|---|---|
| embedding | $E$ | 1.61e-16 |
| attention query matrix | $W_Q^{(1)}$ | 6.17e-190 |
| attention key matrix | $W_K^{(1)}$ | 1.47e-7 |
| attention value matrix | $W_V^{(1)}$ | 1.56e-114 |
| attention query matrix | $W_Q^{(1)}$ | 6.17e-190 |
| attention output matrix | $W_O^{(1)}$ | 0.010 |
| MLP gate projection | $G^{(1)}$ | 0.517 |
| MLP up projection | $U^{(1)}$ | 0.716 |
| MLP down projection | $D^{(1)}$ | 6.03e-80 |

Table 10: $\phi_{U^{(\ell)}}$ on parameters from `StripedHyena-Nous-7B` and `Mistral-7B-v0.1`, some with low p-values.

trained it on the Dolma dataset (Soldaini et al. (2024)) for up to 18B tokens. We trained a second model with independently chosen initialization and data ordering. By only changing initialization and data ordering (i.e. the two main sources of randomness), we have two models that are essentially as similar as two independent models can be.

We keep checkpoints for both seeds after 100M, 1B, 10B, and 18B train tokens. We evaluate the statistics $\phi_{U^{(\ell)}}, \phi_{H^{(\ell)}}$, and $\phi_{\text{MATCH}}$ on the two models at each training checkpoint, reported in Table 9. We highlight that the p-values are broadly distributed, validating our tests can support independence even on two similarly-trained but independent models. We find that all test statistics work well, and there is also little difference in the results at different training checkpoints.

We emphasize that for these experiments, the models are **independent** as the seeds for parameter initialization are manually set to be different. This is in contrast to the MLP retraining experiments, where the models were not independent.

## H ROBUST TEST STATISTIC EXPERIMENTAL RESULTS

### H.1 STRIPED HYENA EXPERIMENTS

We report $\phi_{U^{(\ell)}}$ on specific parameters from `StripedHyena-Nous-7B` and `Mistral-7B-v0.1` shown in Table 10. We no longer only evaluate $\phi_{U^{(\ell)}}$ on MLP up projection matrices, so that we can investigate similarity in other parameters as well. These p-values no longer satisfy the independence requirement of Theorem 2, so we do not aggregate them with `FISHER`.

### H.2 MODEL BLOCK MATCHING

As described in 4.4.2, we can run $\phi_{\text{MATCH}}$ on all pairs of Transformer blocks between two models (of different architecture), as long as they share the GLU structure. In addition to the Llama 3 results, we report results of matched blocks on the Sheared-LLaMa and Nvidia-Minitron models, which are both pruned from Llama models.

In particular, we were able to identify the specific Transformer blocks of $\theta_{8B} = $ `Llama-3.1-8B` whose weights were likely used in initializing $\theta_{3B} = $ `Llama-3.2-3B` and $\theta_{1B} = $ `Llama-3.2-1B`, as Meta reported that the `Llama-3.2-3B` and `Llama-3.2-1B` models were pruned from `Llama-3.1-8B`

(MetaAI (2024)). We use $\phi_{\text{MATCH}}$ on all pairs of MLP blocks, where $(d_{\theta_{8B}}, h_{\theta_{8B}}, N_{\theta_{8B}}) = (4096, 14336, 32), (d_{\theta_{3B}}, h_{\theta_{3B}}, N_{\theta_{3B}}) = (3072, 8192, 28)$, and $(d_{\theta_{1B}}, h_{\theta_{1B}}, N_{\theta_{1B}}) = (2048, 8192, 16)$. We match blocks when the statistic $\phi_{\text{MATCH}}^{(i,j)}$ from block $i$ of model 1 and block $j$ of model 2 is less than 1e-4, reported in Tables 11 and 12 (with the same for the other matchings in this section).

| $i$ | 1 | 2 | 3 | 4 | 5 | 6 | 7 | 8 | 9 | 10 | 11 | 12 | 13 | 14 | 15 | 16 |
|---|---|---|---|---|---|---|---|---|---|---|---|---|---|---|---|---|
| $j : \phi_{\text{MATCH}}^{(i,j)}(\theta_{8B}, \theta_{3B}) < 1e-4$ | 1 | 2 | 3 | 4 | 5 | 6 | 7 | 8 | 9 | 10 | | 11 | 12 | 13 | 14 | 15 |

| $i$ | 17 | 18 | 19 | 20 | 21 | 22 | 23 | 24 | 25 | 26 | 27 | 28 | 29 | 30 | 31 | 32 |
|---|---|---|---|---|---|---|---|---|---|---|---|---|---|---|---|---|
| $j : \phi_{\text{MATCH}}^{(i,j)}(\theta_{8B}, \theta_{3B}) < 1e-4$ | | 16 | 17 | 18 | 19 | 20 | | 21 | 22 | 23 | 24 | 25 | | 26 | 27 | 28 |

Table 11: $\theta_{8B} = $ `Llama-3.1-8B` blocks matched with $\theta_{3B} = $ `Llama-3.2-3B` blocks using $\phi_{\text{MATCH}}$

| $i$ | 1 | 2 | 3 | 4 | 5 | 6 | 7 | 8 | 9 | 10 | 11 | 12 | 13 | 14 | 15 | 16 |
|---|---|---|---|---|---|---|---|---|---|---|---|---|---|---|---|---|
| $j : \phi_{\text{MATCH}}^{(i,j)}(\theta_{8B}, \theta_{1B}) < 1e-4$ | 1 | 2 | 3 | 4 | 5 | 6 | | | 7 | | | 8 | | | 9 | |

| $i$ | 17 | 18 | 19 | 20 | 21 | 22 | 23 | 24 | 25 | 26 | 27 | 28 | 29 | 30 | 31 | 32 |
|---|---|---|---|---|---|---|---|---|---|---|---|---|---|---|---|---|
| $j : \phi_{\text{MATCH}}^{(i,j)}(\theta_{8B}, \theta_{1B}) < 1e-4$ | | 10 | | | 11 | | | | | | | | | | 15 | 16 |

Table 12: $\theta_{8B} = $ `Llama-3.1-8B` blocks matched with $\theta_{1B} = $ `Llama-3.2-1B` blocks using $\phi_{\text{MATCH}}$

Next, we have Sheared-LLaMa 2.7B, with 32 Transformer blocks, hidden dimension 2560 and MLP dimension 6912. All 32 blocks align with the 32 blocks of Llama 2 7B, although both hidden and MLP dimensions have been reduced through pruning.

| $i$ | 1 | 2 | 3 | 4 | 5 | 6 | 7 | 8 | 9 | 10 | 11 | 12 | 13 | 14 | 15 | 16 |
|---|---|---|---|---|---|---|---|---|---|---|---|---|---|---|---|---|
| $j : \phi_{\text{MATCH}}^{(i,j)}(\theta_1, \theta_2) < 1e-90$ | 1 | 2 | 3 | 4 | 5 | 6 | 7 | 8 | 9 | 10 | 11 | 12 | 13 | 14 | 15 | 16 |

| $i$ | 17 | 18 | 19 | 20 | 21 | 22 | 23 | 24 | 25 | 26 | 27 | 28 | 29 | 30 | 31 | 32 |
|---|---|---|---|---|---|---|---|---|---|---|---|---|---|---|---|---|
| $j : \phi_{\text{MATCH}}^{(i,j)}(\theta_1, \theta_2) < 1e-90$ | 17 | 18 | 19 | 20 | 21 | 22 | 23 | 24 | 25 | 26 | 27 | 28 | 29 | 30 | 31 | 32 |

Table 13: $\theta_1 = $ `Sheared-LLaMa 1.3B` blocks matched with $\theta_2 = $ `Llama-2-7B` blocks using $\phi_{\text{MATCH}}$

Next, we have Sheared-LLaMa 1.3B, with 24 Transformer blocks, hidden dimension 2048 and MLP dimension 5504.

| $i$ | 1 | 2 | 3 | 4 | 5 | 6 | 7 | 8 | 9 | 10 | 11 | 12 |
|---|---|---|---|---|---|---|---|---|---|---|---|---|
| $j : \phi_{\text{MATCH}}^{(i,j)}(\theta_1, \theta_2) < 1e-5$ | 1 | 2 | 3 | 4 | 5 | 6 | 7 | 8 | 10 | 12 | | 16 |

| $i$ | 13 | 14 | 15 | 16 | 17 | 18 | 19 | 20 | 21 | 22 | 23 | 24 |
|---|---|---|---|---|---|---|---|---|---|---|---|---|
| $j : \phi_{\text{MATCH}}^{(i,j)}(\theta_1, \theta_2) < 1e-5$ | 17 | 18 | 19 | 20 | 21 | 22 | 25 | 27 | 28 | 29 | 31 | 32 |

Table 14: $\theta_1 = $ `Sheared-LLaMa 1.3B` blocks matched with $\theta_2 = $ `Llama-2-7B` blocks using $\phi_{\text{MATCH}}$

Finally, we compare Llama 3.1 8B with `nvidia/Llama-3.1-Minitron-4B-Depth-Base`, a pruned model by reducing from 32 to 16 Transformer blocks and are able to identify the likely shared blocks.

| $i$ | 1 | 2 | 3 | 4 | 5 | 6 | 7 | 8 | 9 | 10 | 11 | 12 | 13 | 14 | 15 | 16 |
|---|---|---|---|---|---|---|---|---|---|---|---|---|---|---|---|---|
| $j : \phi_{\text{MATCH}}^{(i,j)}(\theta_1, \theta_2) < 1e-90$ | 1 | 2 | 3 | 4 | 5 | 6 | 7 | 8 | 9 | 10 | 11 | 12 | 13 | 14 | 15 | 32 |

Table 15: $\theta_1 = $ `nvidia/Llama-3.1-Minitron-4B-Depth-Base` blocks matched with $\theta_2 = $ `Llama-2-7B` blocks using $\phi_{\text{MATCH}}$

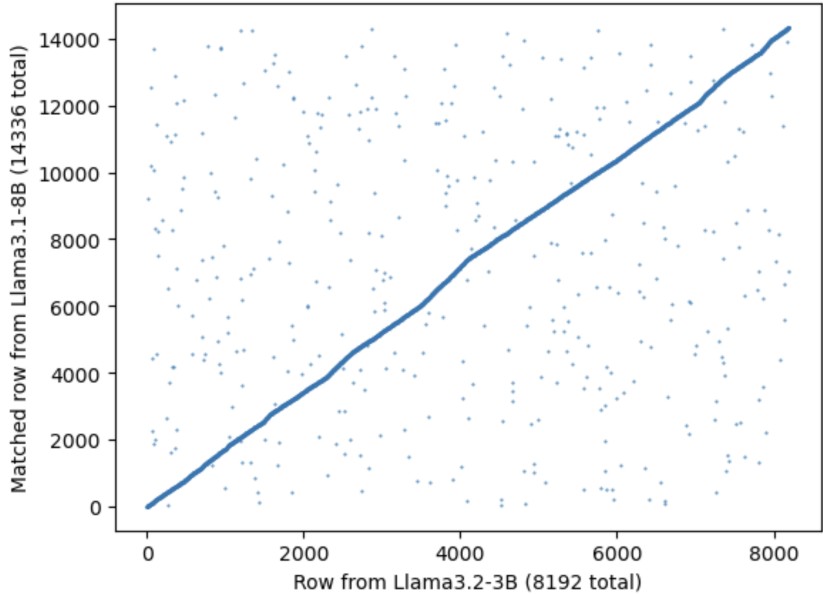

Figure 11: Up projection activations aligned from first MLPs of Llama 3.1 8B and Llama 3.2 3B.

### H.2.1 ACTIVATIONS MATCHING

By using $\phi_{\text{MATCH}}$ on two MLPs from two models, we can examine the permutation $\pi$ returned from just the gate or up projection matching, $\text{MATCH}(H_{\theta_1,\text{up}}^{(\ell)}, H_{\theta_2,\text{up}}^{(\ell)}$ from Section 3.3. This returns which rows of the activation matrices are best aligned. For pruned models where dimensions are reduced, this can provide insight into how activation rows were selected for the smaller MLPs.

The plot in Figure 11 shows the activation rows from the up projection matrix $U^{(1)}$ of the first MLP of Llama 3.2 3B (8192 total) (on the x-axis) matched with the rows from the up projection matrix of the first MLP of Llama 3.1 8B (out of 14336 total) (on the y-axis). In particular, we can see that the activations are not simply the first the first 8192 rows pruned from the 14336-dimensional MLP, rather they appear to be distributed across all 14336 rows.

### H.3 DISTILLING MODELS WITHOUT A GLU MLP

Finally, we show it is possible to extend the robust statistic $\phi_{\text{MATCH}}$ to models that do not have a GLU MLP as well. In particular, we distill any other MLP or feedforward network with a GLU MLP. In principle, one could replace any series of layers mapping inputs to activations with a GLU MLP from Definition 3. We reinitialize the layers with a GLU MLP, $G, U, D$. Then with the same setup as the MLP retraining from Section 4.2.1, we sample inputs and compute the outputs from the original model layers, and minimize MSE loss over the outputs.

We retrain the first MLP of both `manupande21/GPT2_PMC` and `openai-community/gpt2`, where the former is a finetune of the latter (Radford et al. (2019)). These models use a standard 2-layer FFN (Example 1) rather than a GLU MLP. After 10K training steps, we run $\phi_{\text{MATCH}}^{(1)}$ on the first MLP, which yields a p-value of 7.955e-83, showing that $\phi_{\text{MATCH}}$ may be extended to other architectures as well via distilling.

