# OpenReview forum: "Independence Tests for Language Models"
_ICLR.cc/2025/Conference — Submitted to ICLR 2025_

### Official Review · Reviewer_hjpi · 2024-10-18

**Soundness:** 3
**Presentation:** 2
**Contribution:** 3
**Rating:** 6
**Confidence:** 3

**Summary:**

This paper explores the challenge of verifying whether two language models were trained independently or if one was fine-tuned from the other. The authors introduce a family of statistical tests to assess model independence by calculating exact p-values based on different measures of similarity in the models' weights and activations. These tests demonstrate robustness, remaining effective despite variations in training data or adversarial transformations that modify weights without affecting model output. Furthermore, the paper emphasizes the resilience of these tests even after substantial fine-tuning, providing a comprehensive evaluation across multiple models and fine-tuning strategies.

**Strengths:**

1. The paper offers a thorough and insightful examination of the problem, providing a novel approach to verifying model independence.
2. The authors present a rigorous method for testing model independence, incorporating statistical tests that generate valid p-values.
3. The proposed tests have been demonstrated to be effective across a range of models and transformations, exhibiting robustness even in complex scenarios and providing strong results across various experimental conditions.

**Weaknesses:**

1. Experiments mainly use decoder-only architectures, all around 7B parameters, lacking variation in model types and sizes.

2. Techniques like aligning hidden activations could be computationally expensive for larger or more complex models.

3. The tests focus on specific architectures (Llama-2) without exploring how different hyperparameters (e.g., layers, model size) affect results.

**Questions:**

1. Whether the proposed framework remains effective for encoder-decoder architectures (e.g., BERT) or encoder-only models? The framework’s validity across these different architectures is not discussed.
2. How sensitive is the framework to larger models beyond the 7B parameter range, or to smaller-scale models such as Phi-3?
3. Since the framework only detects dependence in open-source models, what are its specific real-world applications? Given the freedom and lack of control over open-source models, the framework may be limited to verifying model provenance in research or regulatory contexts.
4. What is the limitation of this framework?

---

> ### Author Response · Authors · 2024-11-21
> **Response to Reviewer hjpi Part 1**
>
> We thank the reviewer for their feedback.
>
> > **W1/W3** Experiments mainly use decoder-only architectures, all around 7B parameters, lacking variation in model types and sizes.
>
>
> We agree with the reviewer’s concerns. We have since evaluated our different tests on various architectures such as Mistral, BERT, StripedHyena, and Llama3. We have also added experiments on models of different parameter sizes such as Llama 3.2 1B and Llama3.2 3B. We emphasize we are able to use our robust statistic even on two models with **different architecture or dimensions**.
>
> > **Q1** Whether the proposed framework remains effective for encoder-decoder architectures (e.g., BERT) or encoder-only models?
>
> Yes, the p-values from our statistics $\\phi_\\text{CSU}$ and $\\phi_\\text{CSH}$ are valid by construction for any models trained using a $\\Pi$-invariant learning algorithm, such as SGD or Adam, regardless of architecture (encoder or decoder, etc.). Empirically, our tests seem to achieve very high recall for BERT models. In particular, during the rebuttal period we tried running $\\phi_\\text{CSU}$ on the based model google-bert/bert-base-uncased and the fine-tune thereof dima806/tweets-gender-classifier-distilbert, obtaining a p-value < 1e-308.
>
> > **Q2** How sensitive is the framework to larger models beyond the 7B parameter range, or to smaller-scale models such as Phi-3?
>
> Our tests work for different architectures and large parameter models as well. For example, we run $\\phi_\\text{CSU}$ on 70B parameter models of the Llama architecture and verify its effectiveness. In the table below, the test correctly identifies Llama2-70B and Miqu-1-70B as fine-tunes, but not Llama2-70B and Llama3.1-70B.
>
> | $\\theta_1 = $ Llama-2-70b-hf, $\\theta_2 = $? | $\\phi_\\text{CSU}$ |
> | :-------------:    | :--------------: |
> | miqu-1-70b-pytorch | $<\\varepsilon=$1e-308  |
> |  Llama-3.1-70B  |  0.571  |
> |   Palmyra-Fin-70B-32K | 0.539 |
>
>
> Continuing with the non-robust tests, we also evaluate on Mistral models, and find, for example, layers from StripedHyena-Nous-7B that are not independent of layers from Mistral-7B-v0.1, in the table below
>
> | Parameter name | Notation | $\\phi_\\text{CSU}$ |
> | :--: | :--: | :--: |
> | embedding | $E$ | 1.61e-16 |
> | attention query matrix | $W_Q^{(1)}$ | 6.17e-190 |
> | attention key matrix | $W_K^{(1)}$ | 1.47e-7 |
> | attention value matrix | $W_V^{(1)}$ | 1.56e-114 |
> | attention query matrix | $W_Q^{(1)}$ | 6.17e-190 |
> | attention output matrix | $W_O^{(1)}$ | 0.010 |
> | MLP gate projection | $G^{(1)}$ | 0.517 |
> | MLP up projection | $U^{(1)}$ | 0.716 |
> | MLP down projection | $D^{(1)}$ | 6.03e-80 |
>
> The tests also work on smaller-scale models such as Phi-3. Both $\\phi_\\text{CSH}$ and $\\phi_\\text{MATCH}$ return a statistic < 1e-308 (aggregated with Fisher’s method) on the fine-tuned model pair microsoft/Phi-3.5-mini-instruct and numind/NuExtract-v1.5 (3.8B parameters).
>
> Furthermore, the robust test $\\phi_\\text{MATCH}$ can even be used on model pairs of differing architectures. We run the robust MLP matching statistic $\\phi_\\text{match}$ on (all pairs of blocks of) $\\theta_{8B} = $ Llama 3.1-8B and $\\theta_{3B} = $ Llama 3.1-3B, which have very different model dimensions — the number of layers, embedding dimension, and MLP dimension are all different. (This is possible because the LAP element of $\\phi_\\text{match}$ can match a permutation from hidden activation matrices of different shapes — and returns one of the smaller dimension.) We find that the robust statistic returns a value < 1e-4 and can **identify the specific Transformer blocks of Llama 3.1-8B very likely used to initialize Llama 3.1-3B through pruning.**
>
> | $i$ | 1 | 2 | 3 | 4 | 5 | 6 | 7 | 8 | 9 | 10 | 11 | 12 | 13 | 14 | 15 | 16 | 17 | 18 | 19 | 20 | 21 | 22 | 23 | 24 | 25 | 26 | 27 | 28 | 29 | 30 | 31 | 32 |
> | :-: | :-: | :-: | :-: | :-: | :-: | :-: | :-: | :-: | :-: | :-: | :-: | :-: | :-: | :-: | :-: | :-: | :-: | :-: | :-: | :-: | :-: | :-: | :-: | :-: | :-: | :-: | :-: | :-: | :-: | :-: | :-: | :-: |
> $j: \\phi_\\text{MATCH}^{(i,j)}(\\theta_{8B},\\theta_{3B}) < 10^{-4}$ | 1 | 2 | 3 | 4 | 5 | 6 | 7 | 8 | 9 | 10 | | 11 | 12 | 13 | 14 | 15 | | 16 | 17 | 18 | 19 | 20 | | 21 | 22 | 23 | 24 | 25 | | 26 | 27 | 28 |
>
> The flexibility of $\\phi_\\text{match}$ for varying size, and running $\\phi_\\text{MATCH}$ on all pairs of MLP blocks, is significant because it prevents adversaries that may take only certain layers of a pre-trained model and inject other layers (i.e. through pruning).

---

> > ### Author Response · Authors · 2024-11-21
> > **Response to Reviewer hjpi Part 2**
> >
> > > **W2** Techniques like aligning hidden activations could be computationally expensive for larger or more complex models.
> >
> > We understand this concern, and we report computational costs of our tests when run on a Nvidia RTX A6000 GPU: for $\\phi_\\text{CSH}$ on two 7B models, the test on 32 Transformer blocks takes on average less than 2 minutes. Note that this test also involves aligning hidden units (activations or weights) and is not very computationally expensive.
> >
> > (For our robust test, we run $\\phi_\\text{MATCH}$ on two 7B models, the test for 32 Transformer blocks may take up to 5 hours, and on average 5-10 minutes per Transformer block. This includes each forward pass and using LAP to align the hidden activations. The reason this is longer is because we run the forward pass for each Transformer block and also align matrices of dimension $h x nd$, where $h$ is the MLP dimension, $d$ is the embed dimension, and $n$ is the number of input sequences.
> >
> > The runtime of LAP is $O(hdlog(h))$ so for larger models, this will scale only linearly in both dimensions (i.e. on a single Transformer block for a Llama 70B model, will take 4 times as long as a 7B model).
> >
> > In addition, if a model developer is concerned about model theft by a particular developer, they will only need to run this test on a single model pair, and will likely have the computational resources for this.
> >
> > > **Q3**  Since the framework only detects dependence in open-source models, what are its specific real-world applications?
> >
> > Our framework enables third parties to test whether two models were independently trained given their weights. Thus, anyone can currently apply our tests to investigate whether any two open-weight models were independently trained versus not. We agree our tools might find useful application in research and regulatory contexts: consider a hypothetical IP dispute where the provider of a commercial *closed-weight* model is suspected of violating the license of some open-weight model by fine-tuning it for commercial use without attribution. In this hypothetical, in principle a trusted independent third party may be given access to the weights of the closed model (e.g., via a court order) in order to apply our tests. Such IP disputes have already arisen in practice. For example, Reflection—a startup—recently used Llama3-70B as a base model to fine-tune a model that was released under their IP but without crediting Meta. . This is a violation of the Meta license (see link below), which stipulates users who fine-tune Llama3 and use it for commercial purposes must acknowledge the base model name as a prefix.
> >
> > Source(s):
> > https://github.com/meta-llama/llama-models/blob/main/models/llama3/LICENSE
> > https://www.tomsguide.com/ai/the-reflection-70b-model-held-huge-promise-for-ai-but-now-its-creators-are-accused-of-fraud-heres-what-went-wrong
> >
> > > **Q4** What is the limitation of this framework?
> >
> > The main limitation is that we do not distinguish between various “non-independent cases,” such as determining the direction of fine-tuning (was A fine-tuned from B, or was B fine-tuned from A?) or whether two models were independently fine-tuned from a common base model (was A fine-tuned from B, or were A and B both fine-tuned independently from C). Integrating metadata such as a model's release date can help answer such questions, though we leave this direction to future work.

---

> > > ### Comment · Reviewer_hjpi · 2024-12-03
> > >
> > > I would like to thank the authors for their efforts in addressing my concerns in the rebuttal. After reviewing the comments of the other reviewers and the revised manuscript, I believe my concerns have been addressed. Therefore, I am willing to revise my score to 6.

---

### Official Review · Reviewer_dVJG · 2024-11-04

**Soundness:** 3
**Presentation:** 3
**Contribution:** 3
**Rating:** 6
**Confidence:** 3

**Summary:**

The submission presents a statistical test to check whether two language models were independently trained. Notably, the test gives exact p-values. The test involves constructing a set of transforms such that one can generate many independent copies of a model by simply applying randomly sampled transformations. A test statistic can then be applied to the set of transformed models for obtaining a p-value. The submission evaluates a selection of different test-statistics, and some demonstrate very high power.

**Strengths:**

1.	Submission motivates problem well. It indeed seems important to be able to track provenance of open-weight models.
2.	Submission is well-written and explains method clearly.
3.	Submission evaluates method on a variety of existing models and models trained for this evaluation.
4.	Proposed method shows a large improvement over existing baselines, and appears to have meaningfully novel contributions.
5.	Submission considers a certain class of adaptive attacks, and introduces a variation of their method to defend against them.

**Weaknesses:**

1.	Submission does not evaluate methods against "most relevant prior work" (Zeng et al., 2024). Presumably this is because the method is not robust and does not provide exact p-values, however the submission's own methods ($ϕ_{CSU}$ and $ϕ_{CSH}$) are shown to be non-robust, and $ϕ_{JSD}$ does not provide p-values. Yet all three tests are evaluated in Section 4.1. It would strengthen the submission to evaluate against Zeng et al., 2024.
2.	Submission does not describe cost of algorithms. This seems important given that certain decisions are made based on computation cost (using a lookup table on line 266, and using T=99 on line 419).
3.	More clarity could be provided related to some of the simplifying steps when computing p-values. Specifically, in section 3.2.2 when stating “instead of running PERMTEST itself, in our experiments we convert the output $\hat{p}$ to an approximate p-value $\hat{p}$ using standard lookup tables for Spearman correlation coefficient” Please describe more details on why this is possible. Also, does this relate to why the p-values for the dependent models are all epsilon? It’s surprising that every model that is dependent has such a low p-value. Please provide some additional description or intuition for why that is expected.

**Questions:**

1.	Suggestion: evaluate against statistic from Zheng et al., 2024, as mentioned above.
2.	Suggestion: give some estimate of the rough cost to run such a test.
3.	Suggestion: If feasible, running $ϕ_{l2}$ with T > 99 would provide a better comparison to the other methods. In particular, $ϕ_{CSU}$ and $ϕ_{CSH}$ have relatively low p-values for xgen-7b-4k-base, while $ϕ_{l2}$ does not, potentially indicating that $ϕ_{l2}$ would be better that the other methods if given a higher T?
4.	Question: Algorithm 3 uses the independence of blocks to increase the power of the submission's proposed tests. If one calculated $ϕ_{l2}$ using the parameters of only a single block, could that use Algorithm 3 as well to increase the power? That might provide a more comparable baseline. If that is the case, I would suggest evaluate against that as well.
5.	Suggestion: In section 4.3.1, highlight that you are evaluating $ϕ^i_{MATCH}$ rather than $ϕ_{MATCH}$.
6.	Question: Why were the other baselines not evaluated in Table 3?
7.	Suggestion: In Figures 4, 5, 6, and 7, it might be helpful to highlight which model pairs are dependent.
8.	Suggestion: The use of the word 'sequel' (Section 3.2.1 and line 388) is confusing (what is the sequel, and the sequel to what?). I'd suggest clarifying what you mean by sequel or selecting a more appropriate word.

---

> ### Author Response · Authors · 2024-11-21
> **Response to Reviewer dVJG Part 1**
>
> We thank the reviewer for their comments and questions!
>
> > **W1/Q1**: Submission does not evaluate methods against "most relevant prior work" (Zeng et al., 2024)
>
> Thank you for the suggestion. We evaluated the three statistics $M_a$, $M_b$, and $M_c$ proposed by Zeng et al., 2024. In the non-robust setting for non-independent pairs, where the statistics (measure of cosine similarity) are comparable to our findings.
>
> |$\\theta_1 = \\texttt{Llama-2-7b-hf}, \\theta_2=$ | Independent? | $M_a$ | $M_b$ | $M_c$ | $\\phi_\\text{MATCH}$ | $\\phi_\\text{CSU}$ | $\\phi_\\text{CSH}$ | $\\phi_\\text{JSD}$ |
> | :---: | :---: | :---: | :---: | :---: | :---: | :---: | :---: | :---: |
> |$\\texttt{vicuna-7b-v1.5}$ | No | 1.0 | 0.9883 | 0.9922 | $< \\varepsilon$ | $< \\varepsilon$ | $< \\varepsilon$ | -10.874|
> |$\\texttt{Nous-Hermes-llama-2-7b}$ | No | 1.0 | 1.0 | 1.0 | $< \\varepsilon$ | $< \\varepsilon$ | $< \\varepsilon$ | -12.101|
> |$\\texttt{llama-7b-hf}$ | Yes | 0.0884 | 0.0250 | 0.0400 | 0.049 | 0.595 | 0.253 | -11.102|
> |$\\texttt{AmberChat}$ | Yes | 0.1289 | -0.0093 | 0.0198 | 0.941 | 0.460 | 0.279 | -10.281 |
> |$\\texttt{Openllama-v1}$ | Yes | 0.1084 | 0.0076 | 0.0057 | 0.286 | 0.357 | 0.703 | -8.381 |
> |Rotated $\\texttt{Llama-2-7b-hf}$ | No | 0.0767 | 0.0908 | 0.1011  | $< \\varepsilon$ | 0.517 | 0.323 | N\A |
> |Rotated $\\texttt{vicuna-7b-v1.5}$ | No | 0.1553 | 0.0933 | 0.0977 | $< \\varepsilon$ | 0.688 | 0.857 | -10.874 |
> |Rotated $\\texttt{Nous-Hermes-llama-2-7b}$ | No | 0.0332 | 0.0718 | 0.1060 | $< \\varepsilon$ | 0.772 | 0.240 | -12.101|
>
>
> However, we can transform the weights of each model (without changing the output, as we describe in Appendix D), in a way that completely breaks the effectiveness of all of their tests (and our non-robust tests), while our robust test based on $\\phi_\\text{MATCH}$ still yields an astronomically small value.
>
> > **W2/Q2** Submission does not describe cost of algorithms.
>
> Thank you for the suggestion, we will revise our paper. When running the original permutation test, such as using the L2 statistic and T=99, we must permute the model 99 times and compute the L2 distance between the permuted weights and the other model’s weights each time. On a GPU (Nvidia RTX A6000), this takes around 5 minutes. The runtime of this test scales linearly with T—thus, the cost quickly becomes prohibitive if we want to obtain a small p-value (e.g., in order to obtain a p-value < 1e-10 we would need T > 1e10). This trade-off between cost versus power motivates scipy.stats to estimate p-values (see below) when applicable (i.e., for $\\phi_\\text{CSU}$ and $\\phi_\\text{CSH}$) instead of running PERMTEST.
>
> We apologize for the confusing usage of “lookup table”, and will change our writing. Our implementation uses the Spearman’s rank correlation statistic using scipy.stats, which indicates the probability of an uncorrelated system producing datasets that have a Spearman correlation at least as extreme as the one computed from these datasets. Note that this p-value is still valid (up to numerical precision issues) under our null hypothesis.
>
> >**W3** More clarity could be provided related to some of the simplifying steps when computing p-values. Specifically, in section 3.2.2 when stating “instead of running PERMTEST itself, in our experiments we convert the output
>  to an approximate p-value
>  using standard lookup tables for Spearman correlation coefficient”
>
> The reason we can avoid running PERMTEST for the Spearman correlation-based test statistics (e.g., $\\phi_\\text{CSW}$ and $\\phi_\\text{CSH}$) is because the distributions of these test statistics are the same for any two independent models (contrast this with $\\ell_2$ distance, whose distribution will vary for different pairs of independent models depending on factors such as the amount of weight decay employed during training, etc.). Thus, because the null distribution of the Spearman correlation coefficient is known, we do not have to run PERMTEST to keep simulating these null distributions, and we can simply convert the Spearman correlation coefficient output by both $\\phi_\\text{CSU}$ and $\\phi_\\text{CSH}$ to a p-value using the scipy.stat library.
>
> The fact that we do not have to run PERMTEST means we are not computationally bottlenecked like in the case of $\\ell_2$ distance; in particular, in the case of $\\ell_2$ distance we must permute each model more T times to obtain a p-value less than 1/{T+1}. Thus, we are able to report low p-values that more accurately reflect the true statistical power of our tests. We expect in principle that we could obtain similarly low p-values using $\\ell_2$ distance if we were able to run PERMTEST with T > 1/$\\varepsilon$; of course, doing so is computationally infeasible.

---

> > ### Author Response · Authors · 2024-11-21
> > **Response to Reviewer dVJG Part 2**
> >
> > > **Q3/4** Question: Algorithm 3 uses the independence of blocks to increase the power of the submission's proposed tests. If one calculated L2 using the parameters of only a single block, could that use Algorithm 3 as well to increase the power?
> >
> > Thank you for the suggestion! Indeed, this would increase the power of the $\\phi_\\text{L2}$ test, and the independence condition between Transformer blocks still holds under the null. To make the \phi_L2 statistic more aligned with the other statistics, we consider:
> > $\\phi_{\\ell_2}^{(i)} = \\ell_2(\\theta_{\\text{block}, 1}^{(i)}, \\theta_{\\text{block}, 2}^{(i)})$
> > so each $\\phi_{\\ell_2}^{(i)}$ is over only the weights from one Transformer block. Then we can apply Algorithm 3 and use Fisher to aggregate over the L blocks.
> > In the case of our models, where we use T=99 over models with 32 blocks, we get the aggregated p-value of $\\ell_2$ using this method will be Fisher(0.01,...0.01) = 2.55e-31, which is more comparable with the other statistics.
> >
> > The test also has more power than the other baselines like $\\phi_\\text{JSD}$ which does not provide p-values. Hopefully this also resolves the suggestion about using T > 99 for the $\\ell_2$ permutation test, as this gives the test more power.
> >
> > > **Q6** Why were the other baselines not evaluated in Table 3?
> >
> > We didn’t evaluate our test using other statistics like $\\phi_\\text{L2}$ because the test itself is valid by construction. We report $\\phi_\\text{L2}$ and the baseline $\\phi_\\text{JSD}$ here and will add them to the revision as well.
> >
> > | # train tokens | $\\phi_\\text{CSU}$ | $\\phi_\\text{CSH}$ | $\\phi_\\text{L2}$ | $\\phi_\\text{MATCH}$ | $\\phi_\\text{JSD}$ (log) |
> > | :---: | :---: | :---: | :---: | :---: | :---: |
> > | 100M | 0.641 | 0.119 | 0.07 | 0.809 | -11.81 |
> > | 1B | 0.789 | 0.483 | 0.06 | 0.443 | -11.05 |
> > | 10B | 0.707 | 0.277 | 0.93 | 0.343 | -11.28 |
> > | 18B | 0.819 | 0.141 | 0.64 | 0.027 | -11.03 |
> >
> > We find that $\\phi_\\text{L2}$ also identifies the model pairs as independent, as expected. We find again that $\\phi_\\text{JSD}$ is unreliable, as the JSD (exp(-11.81)) between these two independent OLMo models after 18B training tokens is less than the JSD (exp(-10.87)) between the non-independent model pair Llama-2-7b-hf and vicuna-7b-v1.5 (Table 1 in the paper).
> >
> > >**Q5** In section 4.3.1, highlight that you are evaluating \\phi_MATCH^{(i)} rather than \\phi_MATCH.
> >
> > Yes, we are reporting results for $\\phi_\\text{MATCH}^{(i)}$. We will fix this in our revised writing. Thank you for pointing that out.
> >
> > > **Q7** In Figures 4, 5, 6, and 7, it might be helpful to highlight which model pairs are dependent.
> >
> > Thank you for the suggestion. Yes, we will edit these figures in the revision to make the ground truth there more clear.
> >
> > > **Q8**  The use of the word 'sequel' (Section 3.2.1 and line 388) is confusing (what is the sequel, and the sequel to what?).
> >
> > By “sequel” we mean, “from this point on in the paper.” (As in the specific notation applies hereafter). We will use different terminology in the revision.

---

> > > ### Comment · Reviewer_dVJG · 2024-11-25
> > > **Response Follow-Up**
> > >
> > > Thank you for your detailed responses! I will be looking forward to the revised version.

---

### Official Review · Reviewer_nQ6u · 2024-11-04

**Soundness:** 3
**Presentation:** 2
**Contribution:** 2
**Rating:** 3
**Confidence:** 4

**Summary:**

The paper presents a method to estimate if two LLMs are independent of each other, or if their training procedures had co-dependencies. The high-level idea of the proposed strategy is that, if two sets of LLM weights are independent, then the distribution of differences between arbitrary permutations of their weights is uniform. Otherwise, differences for the same permutation (e.g. original weights), will be significantly smaller than the difference amongst arbitrarily permuted versions of the weights from the two models. Through this principle, the methods computes p-values of the distribution of differences, which reflect the probability of the two models being idependent.

**Strengths:**

The paper is mathematically thorough and bases its thesis in well formulated assumptions. The idea is interesting and the method relatively simple and inexpensive, which makes it attractive for implementation.

**Weaknesses:**

1) "Independence" is too loosely defined throughout the paper, and sometimes causes confusion.

For example, in the introduction, in line 60, the authors state "we do not distinguish whether one is a fine-tune of the other or if the two models share a common ancestor". Firstly, this is in contradiction with the first sentence in the abstract: "We consider the following problem of model provenance: can a third party verify whether two language models are trained independently versus fine-tuned from one another given the weights of both models?"

Secondly, the above leads to think that the method can distinguish if two models are somewhat fine-tuned from each other OR from the same model. However, in section 3.1, the independence of the weights depends on both the training algorithm A and initial weights theta^0, which comprehend many more aspects than just common initialisation/fine-tuning from one another. For example, what is the proposed test's output for two models trained on the same data from random initialisation? In this case A_1 and A_2 are not independent; so this would meant that theta_1 and theta_2 are also not independent? What about smaller details, like simply batch size? This would also be a common characteristic of A_1 and A_2. Are the resulting model's weights independent? The definition of statistical independence of equation 1 is not clear and perhaps not entirely proper; it seems to me that the concept of independence in this context is more continuous and might need to be put in terms of co-correlation or similar.

2) The applicability of the method is too restricted in my opinion, for two reasons:

First, because of the problem stated in 1 above; If I get a p-value form the proposed method >0.05, what do I know about the two models I tested? Was one fine-tuned initialising from the other? Did they just have identical training procedure with the same data? Was one fine-tuned with the output of the other in a teacher-student manner? Some sources of co-dependence may be problematic, while others may not, so not knowing where the dependence comes from makes the result difficult to use in practical situations.

Second, the method works only if the two models have exactly the same architecture, while there are many situation in which a model of different architecture is derived from another model. In fact, this is the predominant problem with copyright infringement at present; companies owning proprietary LLMs do not have a problem with models being fine-tuned from their ones as initialisation, as they keep them secret. The problem of intellectual property arises when their outputs, which are publicly accessible, are used to fine-tune another model of different architecture. The proposed method cannot detect these cases, even with access to the proprietary model.

UPDATED REVIEW FOLLOWING REPLIES:

Following the replies and discussion, I am fairly certain that the authors are indeed misunderstanding the post-processing theorem and this is causing the basic starting point of their proposed technique to be problematic. I will try to be clearer, starting from my example and the response referencing it from the Authors.

From the authors previous response: "Thus, in your example, the mutual information between $A_1(\theta_1^0)$  and $A_2(\theta_2^0)$ is zero, which is consistent with our claims."

This is not true. The authors are confusing two random variables being the same random variable, in which case the mutual information is indeed 0, with two different random variable having the same value when observed, which is the case in my example, and for which the mutual information is not 0 (or not necessarily). Let me go through the details:

$I(A_1(\theta_1^0), A_2(\theta_2^0)) = H(p(A_1(\theta_1^0), A_2(\theta_2^0))) - H(p(A_1(\theta_1^0), A_2(\theta_2^0)) | p(A_1(\theta_1^0)), p(A_2(\theta_2^0)))$

Going through the components in the equation above:

$p(A_1(\theta_1^0) =  \int p(\theta_0)p(A)p(A_1(\theta_1^0)|A, \theta_0)dAd\theta_0$ - this is the probability to observe the value $A_1(\theta_1^0)$ alone.

$p(A_2(\theta_2^0) =  \int p(\theta_0)p(A)p(A_2(\theta_2^0)|A, \theta_0)dAd\theta_0$ - similarly the probability to observe the value $A_2(\theta_2^0)$ alone.

$p(A_1(\theta_1^0), A_2(\theta_2^0)) = \int p(A, A') p(\theta_0, \theta_0') p(A_1(\theta_1^0), A_2(\theta_2^0)|A, A', \theta_0, \theta_0')$ - this is the joint probability of observing both $A_1(\theta_1^0)$ and $A_2(\theta_2^0)$.

In my example, I stated that $\theta_1^0 \perp \theta_2^0$, so $p(\theta_0, \theta_0') = p(\theta_0) p(\theta_0')$. However, the mutual information is 0, only if the two As are uncorrelated, i.e., $p(A, A') = p(A) p(A')$, but this is not a given. For example, we could have $p(A, A') = p(A')\delta(A=A')$, which is consistent with my example and for which the mutual information would be positive, unless all $A$ and $A'$ are equal to $A_1$, in which case $p(A')$ is also a delta function (all As in the world are the same A=A_1).

Generalising to the statement of the post-processing inequality, $I(\theta_1^0, \theta_2^0) \geq I(A_1(\theta_1^0), A_2(\theta_2^0))$ only if we know that $A_1$ and $A_2$ are uncorrelated ($p(A, A') = p(A) p(A')$ above), which is an assumption that we cannot make; the two models may have been trained copying each others training procedure or using the same training data, with unknown effects on the correlation of the outcomes.

I apologise for confusing the p-values trend in my final question, but my concern stands; if you are basing the guarantee that your method can detect relation of initialised weights on the inequality $I(\theta_1^0, \theta_1^0) \geq I(A_1(\theta_1^0), A_2(\theta_2^0))$ for any arbitrary two $A_1$ and $A_2$, this is not true in general and does not derive from the post-processing inequality.

I confirm my score, as I believe the paper does contain interesting findings, but bases its theory on an incorrect (or at least still to be proven) assumption, which would need to be addressed specifically through theoretical proof/approximation or extensive experimental evaluation.

**Questions:**

See weaknesses.

---

> ### Author Response · Authors · 2024-11-21
> **Response to Reviewer nQ6u Part 1**
>
> We thank the reviewer for their feedback and questions.
>
> > **W1** For example, in the introduction, in line 60, the authors state "we do not distinguish whether one is a fine-tune of the other or if the two models share a common ancestor"
>
> Thanks for bringing this to our attention, we apologize for the lack of clarity in our writing and will revise the paper. Indeed, the first line should say: "We consider the following problem of model provenance: can a third party verify whether two language models are trained independently or not?”.
>
> > **W1**  "Independence" is too loosely defined throughout the paper, and sometimes causes confusion.
>
> We define independence formally in Section 3.1. We will update the revision to more explicitly refer to this formal definition at other points in the paper (e.g., the Introduction).
>
> > **W1** First, because of the problem stated in 1 above; If I get a p-value form the proposed method >0.05, what do I know about the two models I tested?
>
> A p-value of 0.05 means that if the two models were independently trained, then the probability of  obtaining a p-value at most this large is equal to 0.05, over the randomness of the training algorithms and initializations used to produce both models (so long as one model satisfies our assumption of being the product of a permutation equivariant learning algorithm initialized from a permutation invariant distribution). In general, low p-values provide quantitative evidence that two models are not in fact independent as defined in Section 3.1
>
> > **W1** Some sources of co-dependence may be problematic, while others may not, so not knowing where the dependence comes from makes the result difficult to use in practical situations. Secondly, the above leads to think that the method can distinguish if two models are somewhat fine-tuned from each other OR from the same model. Was one fine-tuned initialising from the other? Did they just have identical training procedure with the same data? Was one fine-tuned with the output of the other in a teacher-student manner?
>
>
> We thank the reviewer for their comments and once again emphasize that we will improve the clarity of our writing. Our tests cannot distinguish whether two models are fine-tuned from each other versus from the same model. On the other hand, even if two models are trained with identical procedures on the exact same data, so long as they are trained independently from independent random initializations our tests will correctly identify them as independent. In the case where one model was fine-tuned using the outputs of another model via teacher-student distillation, our tests will incorrectly identify the models as independent. Crucially, this does *not* mean our tests are invalid: the definition of validity is that the p-values from our tests are uniformly distributed between 0 and 1 under the null hypothesis, which is still the case. Rather, it means there are cases of dependent models which we will not be able to reliably distinguish. Another example of such a case would be if someone independently trained a model but then copied over a single parameter from another model at the end of training; technically the two models would not be independent of each other, yet none of our tests would be able to determine this correctly.
>
> We state in the paper that we only give a p-value for the hypothesis test of whether two model weights are independent if the test is rejected, we do not provide any causal information (one model is fine-tuned from the other), although we can provide more granular information on which layers of the model are dependent on which other layers. Because the p-values we obtain for dependent model pairs are all astronomically low, essentially any threshold will obtain perfect recall (e.g. a p-value threshold of 1e-10 will result in zero false negative errors—at least based on our experiments— and a false negative error rate of exactly 1e-10—due to the guarantees of our method). We will clarify this point in the revision.
>
> Our work is focused on inferring independence purely from parameters. And, we do not treat data as a random variable, so we infer nothing about if the same data was used (which would be especially tricky as most language models likely overlap in some training data from the web). Hence, we only test for independence as defined above, when one model uses another’s weights.

---

> > ### Author Response · Authors · 2024-11-21
> > **Response to Reviewer nQ6u Part 2**
> >
> > >**W2** Second, the method works only if the two models have exactly the same architecture, while there are many situation in which a model of different architecture is derived from another model.
> >
> > We wish to clarify that our robust test actually works for models that do not have the same architecture, and in fact allows fine-grained analyses such as identifying shared Transformer blocks and even shared activations of two models with gated linear units (i.e., GLU MLPs), even if the number of units differs between the two models.
> >
> > We run the robust MLP matching statistic $\\phi_\\text{MATCH}$ on (all pairs of blocks of) Llama 3.1-8B and Llama 3.1-3B, which have very different model dimensions — the number of layers, embedding dimension, and MLP dimension are all different. (This is possible because the LAP element of \phi_robust can match a permutation from hidden activation matrices of different shapes — and returns one of the smaller dimension.) We find that the robust statistic returns a value < 1e-4 and can **identify the specific Transformer blocks of Llama 3.1-8B very likely used to initialize Llama 3.1-3B through pruning.**
> >
> > | $i$ | 1 | 2 | 3 | 4 | 5 | 6 | 7 | 8 | 9 | 10 | 11 | 12 | 13 | 14 | 15 | 16 | 17 | 18 | 19 | 20 | 21 | 22 | 23 | 24 | 25 | 26 | 27 | 28 | 29 | 30 | 31 | 32 |
> > | :-: | :-: | :-: | :-: | :-: | :-: | :-: | :-: | :-: | :-: | :-: | :-: | :-: | :-: | :-: | :-: | :-: | :-: | :-: | :-: | :-: | :-: | :-: | :-: | :-: | :-: | :-: | :-: | :-: | :-: | :-: | :-: | :-: |
> > $j: \\phi_\\text{MATCH}^{(i,j)}(\\theta_{8B},\\theta_{3B}) < 10^{-4}$ | 1 | 2 | 3 | 4 | 5 | 6 | 7 | 8 | 9 | 10 | | 11 | 12 | 13 | 14 | 15 | | 16 | 17 | 18 | 19 | 20 | | 21 | 22 | 23 | 24 | 25 | | 26 | 27 | 28 |
> >
> > The flexibility of $\\phi_\\text{MATCH}$ for varying size, and running $\\phi_\\text{MATCH}$ on all pairs of MLP blocks, is significant because it allows detecting dependence between model parameters even if only a subset of the model parameters are used (i.e. through pruning).
> >
> > > **W2** In fact, this is the predominant problem with copyright infringement at present; companies owning proprietary LLMs do not have a problem with models being fine-tuned from their ones as initialisation, as they keep them secret. The problem of intellectual property arises when their outputs, which are publicly accessible, are used to fine-tune another model of different architecture. The proposed method cannot detect these cases, even with access to the proprietary model.
> >
> > Finally, we agree that the ability to detect which models fine-tune on the outputs from protected proprietary models is a salient topic that we do not address. However, intellectual property considerations can be quite nuanced and we believe our method addresses an important use case.
> >
> > For example, Reflection—a startup—recently used Llama3-70B as a base model to fine-tune a model that was released under their IP but without crediting Meta. . This is a violation of the Meta license (see link below), which stipulates users who fine-tune Llama3 and use it for commercial purposes must acknowledge the base model name as a prefix.  There is also a growing set of companies building open-weight models such as IBM and Mistral, and we believe that these tests will be beneficial and of interest to these open-weight developers if they adopt similar licensing approaches.
> >
> > Source(s):
> > https://github.com/meta-llama/llama-models/blob/main/models/llama3/LICENSE
> > https://www.tomsguide.com/ai/the-reflection-70b-model-held-huge-promise-for-ai-but-now-its-creators-are-accused-of-fraud-heres-what-went-wrong

---

> > ### Comment · Reviewer_nQ6u · 2024-11-25
> > **Independence Definition**
> >
> > I thank the Authors for their clarifications. My core concern remains: I believe the starting definition of independence is still not well defined, especially in the formal definition in section 3.1.
> >
> > It is stated in equation 1 that the proposed test seeks to find out if $\theta_1 \perp \theta_2$, where $\theta_1 \sim A_1(\theta_1^0)$ and $\theta_2 \sim A_2(\theta_2^0)$. There are two issues with this:
> >
> > - what do the authors exactly mean by $\perp$ in this context; ~0 mutual information perhaps?
> >
> > - whatever is the definition of $\perp$, from the explanations above, the authors suggest that if $\theta_1^0 \notperp \theta_2^0$ (not independent inizialization) then $A_1(\theta_1^0) \notperp A_2(\theta_2^0)$ (not independent final weights), but that if $A_1 \notperp A_2$, or even $A_1 = A_2$ (not independent or equal training procedure), then $A_1(\theta_1^0) \perp A_2(\theta_2^0)$, (independent final weights) with no further justification.

---

> ### Author Response · Authors · 2024-11-26
> **Clarifying independence definition**
>
> > what do the authors exactly mean by $\perp$ in this context; ~0 mutual information perhaps?
>
> The notation $\perp$ denotes independence of random variables: $X \perp Y$ means that $P(X = x | Y = y) = P(X = x)$ for all $x,y$. Two random variables have zero mutual information if and only if they are independent, so your interpretation of $\perp$ as implying zero mutual information is correct. See [1] for further clarification regarding the $\perp$ notation.
>
> > whatever is the definition of $\perp$, from the explanations above, the authors suggest that if $\theta_1^0 \notperp \theta_2^0$ (not independent inizialization) then $A_1(\theta_1^0) \notperp A_2(\theta_2^0)$ (not independent final weights)
>
> There are cases where this implication is not true. For example, both $A_1$ and $A_2$ could ignore their inputs and output a weight vector with i.i.d. Gaussian elements. In this case, we would have $A_1(\theta_1^0) \perp A_2(\theta_2^0)$ regardless of the (joint) distribution of $\theta_1^0$ and $\theta_2^0$. On the other hand, for most learning algorithms people use in practice, we generally expect the implication to be true; we say as much in the paper (Line 137-38) mainly to provide the reader with helpful intuition, but we are happy to remove this language if it is misleading.
>
> > $A_1 \notperp A_2$, or even $A_1 = A_2$ (not independent or equal training procedure), then $A_1(\theta_1^0) \perp A_2(\theta_2^0)$, (independent final weights) with no further justification.
>
> In our notation, both $A_1$ and $A_2$ are functions that map an input parameter to an output distribution over parameters. Crucially, we do not consider $A_1$ and $A_2$ random variables, so statements of the form  $A_1 \notperp A_2$ are invalid (in our notation) since $\perp$ and $\notperp$ apply only to random variables. Because $A_1$ and $A_2$ are functions, by the post-processing inequality their outputs must be independent if their inputs are independent (i.e., mutual information can only decrease, so if the inputs have zero mutual information then so must the outputs, which in turn implies independence of the outputs). Independence of the output distributions of $A_1$ and $A_2$ immediately implies independence of samples from these distributions.
>
> [1] https://en.wikipedia.org/wiki/Independence_(probability_theory)

---

> ### Comment · Reviewer_nQ6u · 2024-11-27
> **Response on Independence Definition**
>
> The authors may be misunderstanding the post-processing inequality in their setting, which leads to the significant assumption not currently stated or discussed I have concerns with.
>
> The post-processing inequality states that the mutual information through a process between the variable *before and after the process* can only decrease. In the paper's setting this means that $I(\theta_1^0, \theta_2^0) \geq I(\theta_1^0, A_2(\theta^0_2))$ and $I(\theta_1^0, \theta_2^0) \geq I(A_1(\theta_1^0), \theta_2^0)$. This *does not* imply that $I(\theta_1^0, \theta_2^0) \geq I(A_1(\theta^0_1), A_2(\theta^0_2))$. In fact, we can make a simple edge case to disproof the latter inequality:
> Consider the edge case of $A_1()$ and $A_2()$ being both functions that, regardless of the input, return the same set of weights $\theta_1$. Even if the inputs $\theta_1^0 \perp \theta_2^0$, the outputs $A_1(\theta^0_1) = A_2(\theta^0_2) = \theta_1$ are the same, hence the mutual information between them has increased, going from 0 to being maximal.
>
> The unstated assumption in the paper is that the independence of the final weights $\theta_1 \perp \theta_2$ depends *only* on the independence of their initialisation $\theta_1^0 \perp \theta_2^0$ and *not* on any other training factor, such as training data, learning parameters, etc., which the authors embed in $A_1$ and $A_2$. This is a very strong statement and, as explained above, it is *not* a consequence of the post-processing inequality. This would need to be proven in the theory, or strongly demonstrated experimentally to be accepted as an assumption. In the current paper, it is not stated, but implicitly assumed.
>
> In practice, the validity of this assumption is crucial for the proposed test to be taken as viable proof of relationship of initialisation (common initialisation or parents of one-another). I.e., if I get a high p-value, can I conclude that the initialisation of the two models is related, or is this caused by, e.g., the same fine-tuning data. Vice-versa, if I get a low p-value, can I conclude definitively that the initialisations were independent? Or could it be that they still had the same initialisation, but were trained on different data?

---

> > ### Author Response · Authors · 2024-12-01
> > **Follow up Response for Independence Definition**
> >
> > >The authors may be misunderstanding the post-processing inequality in their setting, which leads to the significant assumption not currently stated or discussed I have concerns with.
> > The post-processing inequality states that the mutual information through a process \textbf{between the variable before and after the process can only decrease}. In the paper's setting this means that $I(\theta_1^0, \theta_2^0) \geq I(\theta_1^0, A_2(\theta^0_2))$ and $I(\theta_1^0, \theta_2^0) \geq I(A_1(\theta_1^0), \theta_2^0)$. This \textbf{does not} imply that $I(\theta_1^0, \theta_2^0) \geq I(A_1(\theta^0_1), A_2(\theta^0_2))$. In fact, we can make a simple edge case to disproof the latter inequality:
> > Consider the edge case of $A_1()$ and $A_2()$ being both functions that, regardless of the input, return the same set of weights $\theta_1$. Even if the inputs $\theta_1^0 \perp \theta_2^0$, the outputs $A_1(\theta^0_1) = A_2(\theta^0_2) = \theta_1$ are the same, hence the mutual information between them has increased, going from 0 to being maximal.
> >
> > Thank you for engaging and for the discussion! The edge case you mention is a useful illustrative example to work through. For any random variable $X$ such that $P(X = x) = 1$ for some $x$, the mutual information between $X$ and itself is zero (i.e., $I(X;X) = 0$). This follows from the definition of mutual information as the difference between entropy and conditional entropy. In particular, we have $I(X;X) = H(X) - H(X | X) = 0$, where the last equality follows from the fact that mutual information is nonnegative and $H(X) = 0$ (since $X$ is deterministic). Thus, in your example, the mutual information between $A_1(\theta^0_1)$ and $A_2(\theta^0_2)$ is zero, which is consistent with our claims.
> >
> >
> > > The unstated assumption in the paper is that the independence of the final weights $\theta_1 \perp \theta_2$ depends \textbf{only} on the independence of their initialisation $\theta_1^0 \perp \theta_2^0$ and \textbf{not} on any other training factor, such as training data, learning parameters, etc., which the authors embed in $A_1$ and $A_2$. This is a very strong statement and, as explained above, it is \textbf{not} a consequence of the post-processing inequality. This would need to be proven in the theory, or strongly demonstrated experimentally to be accepted as an assumption. In the current paper, it is not stated, but implicitly assumed.
> >
> > We prove PERMTEST yields a valid p-value in Theorem 1. We explicitly state all our assumptions in the statement of the theorem. If there is an issue with our proof, we are happy to make corrections or clarifications.
> >
> > > In practice, the validity of this assumption is crucial for the proposed test to be taken as viable proof of relationship of initialisation (common initialisation or parents of one-another). I.e., if I get a high p-value, can I conclude that the initialisation of the two models is related, or is this caused by, e.g., the same fine-tuning data. Vice-versa, if I get a low p-value, can I conclude definitively that the initialisations were independent? Or could it be that they still had the same initialisation, but were trained on different data?
> >
> > First, low p-values (i.e., close to zero) indicate two models are "related" (specifically, not independent) whereas high p-values indicate independence. If you get a low p-value, then you can indeed conclude (under the assumptions that we explicitly describe in the statement of Theorem 1) that the initialization of the two models is not independent (i.e., is related). However, a high p-value does not allow you to conclude that the initializations were independent. Consider a similar example to the edge case you mentioned above, wherein the initializations are dependent but both $A_1$ and $A_2$ deterministically output some parameter $\theta$  and assuming $A_1$ and $A_2$ satisfy the conditions of $\Pi$ equivariance as written in Definition 2 in our paper . In this example, our test would correctly produce a high p-value (i.e., uniform between 0 and 1) since the two models are independent despite the fact that the initializations are not independent.

---

### Official Review · Reviewer_CMeE · 2024-11-06

**Soundness:** 2
**Presentation:** 4
**Contribution:** 3
**Rating:** 6
**Confidence:** 3

**Summary:**

The paper proposes a family of independent tests for assessing whether language models are trained independently, based on a permutation test. Leveraging permutation invariance and equivariance in MLP neurons, it provides exact p-values. Extensive evaluations on 21 open-weight models show that the test performs effectively. Specifically, the framework makes the previously proposed $l_2$ distance more stable. The paper also introduces a robust metric to improve the robustness under adversary, validated using retraining of the MLP.

**Strengths:**

- The paper is well-written, with clear logic in deriving each component of the test.
- The method is novel and addresses an important question.
- Extensive experiments consistently demonstrate the method's effectiveness across various settings.

**Weaknesses:**

- The notation of matrices in Algorithm 2 could be improved, as it currently overlaps with the algorithm notation.
- It would be helpful to include an evaluation of the $l_2$, CSU, and CSH tests under adversary, the MLP retraining.

**Questions:**

- The models examined are standard, trained or fine-tuned dense LLMs. How might the proposed test perform with other architectures, such as mixture-of-experts models? How to properly apply the test when comparing a mixture model and a dense model? If sparsification or pruning methods are applied to create mixtures or sparse models, could that impact the test? Additionally, could an adversary leverage these techniques to circumvent the test?

---

After reading other reviewer's reviews and the author's rebuttal, the reviewer partly agree with Reviewer nQ6u and the AC that the independence assumption in the paper is too strong that the type-1 and type-2 error of $\theta^0_1$ and $\theta^0_2$ should explained in greater detail. The reviewer updated the score based on it.

---

> ### Author Response · Authors · 2024-11-21
> **Response to Reviewer CMeE**
>
> We thank the reviewer for the feedback!
>
> > **W1**: The notation of matrices in Algorithm 2 could be improved, as it currently overlaps with the algorithm notation.
>
> Yes, we will change $A_1$ and $A_2$ in Algorithm 2 to be $W_1$ and $W_2$ instead so there is no overlap. Thank you for pointing that out.
>
> >**W2**: It would be helpful to include an evaluation of the L2, CSU, and CSH tests under adversary, the MLP retraining.
> Yes, we will report those statistics under MLP retraining as well.:
>
> | $\\phi_\\text{MATCH}$ | $\\phi_\\text{CSU}$ | $\\phi_\\text{CSH}$ | $\\phi_\\text{L2}$ |
> | :---: | :---: | :---: | :---: |
> | 2.2e-308 | 0.756 | 0.016 | 0.62 |
>
> For the table, we retrain the first MLP block according to the setup in 4.3.1, and evaluate each statistic only on the layers of the first MLP. We also find that $\\phi_\\text{MATCH}$ is robust to retraining.
>
> >**Q1**:  The models examined are standard, trained or fine-tuned dense LLMs. How might the proposed test perform with other architectures, such as mixture-of-experts models?
>
> Thank you for the question on mixture-of-experts models. We wish to clarify that our model is just as valid to apply to mixture of experts models and that our non-robust tests are valid by construction regardless of model architecture. For the activation tests (including $\\phi_\\text{MATCH}$), we can choose to only compare the hidden activation matrices for one expert MLP, for example.
>
> Regarding the potential problem of sparse models, we find that there is still signal from activation and weight matching. For example, comparing the sparse (89.32% sparsity) SparseLLM/prosparse-llama-2-7b with Llama 2 7B using $\\phi_\\text{CSH}$ gives a p-value less than $\\varepsilon$ = 1e-308 (for all 32 blocks). And the robust $\\phi_\\text{MATCH}$ gives a p-value less than 1e-05 for at least the first 20 MLPs.

---

> > ### Comment · Reviewer_CMeE · 2024-11-22
> >
> > Thanks the author for the response.
> >
> > I would like to further inquire about the mixture-of-experts models. In these models, there are multiple FFN modules. Regarding the test that uses the MLP, the author writes, "we can choose to only compare the hidden activation matrices for one expert MLP." However, if only one set of expert MLPs is dependent, there is only a probability of $n^L$ of selecting this specific set, where $n$ is the number of experts, and $L$ is the number of FFNs. If we include even one matrix that belongs to an independent expert in the comparison, the test will incorrectly indicate independence. This suggests that the test is highly fragile and potentially incapable of reliably identifying dependence in this context.

---

> > > ### Author Response · Authors · 2024-11-23
> > >
> > > Thank you for the comment. We can test a pair of MoE models by applying either $\\phi_\\text{CSU}$ or $\\phi_\\text{CSW}$ to each of the $O(n^L)$ pairs of MLP layers between the two models. Under the null hypothesis that the two MoE models are independent, we can union bound the probability of obtaining a minimum p-value of at most $x$ for $x$ in [0,1] over the $O(n^L)$ tests by $1-(1-x)^{n^L}$. This gives us a valid upper bound on the probability of a false positive error, thus ensuring the test will be valid. In the case that only two of the tested MLPs are dependent, even with only one very small statistic $x$, the bound $1-(1-x)^{n^L}$ will still be small.

---

### Author Response · Authors · 2024-11-21
**Response to all Reviewers**

We thank all the reviewers for their feedback and effort. We appreciate the insights and would like to clarify some points we made in the paper that we will further explain in our writing.

First, we emphasize that while most of our experiments are with models all sharing the LLama2 7B architecture, our tests apply in principle to any model pair for which there exists a pair of layers sharing a common architecture or even a pair of tensors sharing a common shape;
notably, using our tests we are able to confirm that the leaked Miqu 70B model from Mistral derives from Llama 2 70B. We also ran additional experiments that identify which layers of Llama 3.1 8B are incorporated into Llama 3.2 1B and 3B, despite none of these models sharing the same architecture.

Second, we will add exposition to the revision clarifying our problem formulation—in particular, clarifying our null hypothesis. To summarize, our null hypothesis is that the two models were *both* independently initialized and trained; notably, we consider two models to not be independent even if both were independently trained from the same initialization (e.g., two different fine-tunes of Llama 2 7B).

Finally, we have replied to each reviewer individually to address their questions, alongside relevant additional experiments. Thank you again for your time and effort in reviewing our submission

---

### Author Response · Authors · 2024-11-30
**Rebuttal Follow-up to Reviewers**

We thank the reviewers for their comments and for engaging in discussion. We have uploaded our revised paper that addresses these comments and concerns. Sections where we made changes and additions from the original submission are marked in blue. Notably, we introduced simpler notation throughout Section 3 (Methods) for our statistics, which we also hope will make clear that these tests extend beyond Llama models.

We also include a new section (4.3) with results for different architectures including larger and smaller models, as well as models with varying hidden dimensions. We are happy to discuss further.

---

### Meta-Review · Area_Chair_UVdm · 2024-12-21

**Metareview:**

This paper explores an interesting and important topic of testing whether two models are trained "independently". The authors proposed a permutation test to compute the p-value of the null hypothesis that a given pair of model weights are independent, and provided extensive experiments over 210 pairs of models to validate the proposed test. The paper has received extensive discussion both between authors and reviewers, and between reviewers and the AC. I've read the major part of the paper myself, the authors' response, as well as the discussion between the authors and the reviewers.

One of the key claim made in the paper, as evident from Line 138-139, is the following statement:
> Indeed, in practice we expect $H_0$ to obtain whenever $\theta_1$ and $\theta_2$ had independent random initializations, i.e., when $\theta_1^0 \perp \theta_2^0$

Regarding the above claim, I agree with Reviewer nQ6u that this claim is a bit too strong, and does not hold even in the context of fine-tuning. To be more specific, the post-processing inequality $I(\theta_1^0; \theta_2^0) \geq I(A_1(\theta_1^0); A_2(\theta_2^0))$ holds only if both procedures $A_1$ and $A_2$ are deterministic or the randomness in $A_1$ and $A_2$ are also independent of everything else, including both $\theta_1^0$ and $\theta_2^0$. In the setting considered in the paper, both $A_1$ and $A_2$ can be fine-tuning algorithms applied to the same data, which makes them random variables as well that are not independent. For example, imagine $A_1$ and $A_2$ just ignore the given input model parameters and both of them output the mean of the same data $\mathcal{D}$ shared by them. Clearly, in this case we have $I(A_1(\theta_1^0); A_2(\theta_2^0)) > 0$ even if $I(\theta_1^0; \theta_2^0) = 0$. Hence the key implication $\theta_1^0 \perp \theta_2^0 \implies A_1(\theta_1^0) \perp A_2(\theta_2^0)$ does not hold anymore. Note that in this example neither $\theta_1^0$ or $\theta_2^0$ is a fine-tuned model of the other. This scenario has not been properly considered in the paper, either.

Why are such counter-examples important? This means that the null hypothesis is too weak, potentially rendering the test less meaningful or informative, and rejecting it does not provide any substantial insight. I'd also suggest the authors to clearly define the meaning of "two models are trained independently", since the training process is very complicated and involves other nuanced factors such as shared data or information. It seems to me what the authors actually meant is that the initial model parameters $\theta_1^0$ and $\theta_2^0$ are statistically independent.

Given the issues above, I recommend rejection at this point but encourage the authors to incorporate the comments above to further strengthen the paper since indeed this direction is very interesting and practical.

**Additional Comments On Reviewer Discussion:**

There are extensive reviews and at the end the reviewers still didn't converge to a consensus. I read the paper in detail and take a pass of the discussion. I recommend rejection due to technical issues in the current version of the paper. Please see my meta-review for more details.

---

### Decision · Program_Chairs · 2025-01-22

Reject